



**Monitoring of glacier albedo from optical remote-sensing data:**
**application to seasonal and annual surface mass balances**
**quantification in the French Alps for the 2000-2015 period**
Lucas Davaze[1], Antoine Rabatel[1], Yves Arnaud[1], Pascal Sirguey[2], Delphine Six[1], Anne
Letreguilly[1], Marie Dumont[3]
[1] Université Grenoble Alpes, CNRS, IRD, Grenoble INP, IGE, F- 38000 Grenoble, France
[2] National School of Surveying, University of Otago, Dunedin, New Zealand
[3] Météo France, CNRS, CNRM – UMR3589, CEN, F-38000 Grenoble, France
*Correspondence to:* L. Davaze (lucas.davaze@univ-grenoble-alpes.fr)



**Abstract.**
Less than 0.25% of the 250,000 glaciers inventoried in the Randolph Glacier Inventory (RGI
V.5) are currently monitored with in situ measurements of surface mass balance. Increasing
this archive is very challenging, especially using time-consuming methods based on in situ
measurements, and complementary methods are required to quantify the surface mass balance
of unmonitored glaciers. The current study relies on the so-called albedo method, based on the
analysis of albedo maps retrieved from optical satellite imagery acquired since 2000 by the
MODIS sensor, onboard of TERRA satellite. Recent studies revealed substantial relationships
between summer minimum glacier-wide surface albedo and annual surface mass balance,
because this minimum surface albedo is directly related to the accumulation-area ratio and the
equilibrium-line altitude.
On the basis of 30 glaciers located in the French Alps where annual surface mass balance are
available, our study conducted on the period 2000-2015 confirms the robustness and reliability
of the relationship between the summer minimum surface albedo and the annual surface mass
balance. At the seasonal scale, the integrated summer surface albedo is significantly correlated
with the summer surface mass balance of the six glaciers seasonally monitored. For the winter
season, four of the six glaciers showed a significant correlation when linking the winter
surface mass balance and the integrated winter surface albedo, using glacier-dependent
thresholds to filter the albedo signal (threshold from 0.53 to 0.76). These results are promising
to monitor both annual and seasonal glacier-wide surface mass balances of individual glaciers
at a regional scale using optical satellite images. A sensitivity study on the computed cloud
masks revealed a high confidence in the retrieved albedo maps, restricting the number of
omission errors. Albedo retrieval artifacts have been detected for topographically incised
glaciers, highlighting limitations in the shadows correction algorithm, although inter-annual
comparisons are not affected by systematic errors.



## 1  Introduction


Mountain glaciers represent only 3% of the ice volume on the Earth but contribute
significantly to sea level rise (Church et al., 2013; Ohmura, 2004). In addition, millions of
people partly rely on glaciers, either for drinking water, agriculture or related glacier hazards
(Chen and Ohmura, 1990; Immerzeel et al., 2010; Kaser et al., 2010). The surface mass
balance (SMB) of glaciers is directly driven by the climate conditions; consequently, glaciers
are among the most visible proxies of climate change (Stocker et al., 2013). Measuring and
reconstructing glacier SMB therefore provides critical insights on climate change both at
global and regional scales.
Systematic SMB monitoring programs began in the late 1940s - early 1950s in most of the
European countries (e.g., Sweden, France, Switzerland, Norway). Gradually, more glaciers
have become monitored, reaching the present worldwide figure of 440. However, this
represents only a little sample of the nearly 250,000 inventoried glaciers worldwide (Pfeffer et
al., 2014). Among the existing methods to quantify changes in glacier SMB, the well-
established glaciological method has become a standard widely used worldwide yielding most
of the reference datasets (World Glacier Monitoring Service, WGMS, Zemp et al., 2015).
Based on repeated in situ measurements, this method requires intensive fieldwork. This
method is however unable to reconstruct SMB of unmonitored glaciers. The Global Terrestrial
Network for Glaciers (GTN-G) aims at increasing substantially the number of monitored
glaciers to study regional climate signal through changes in SMB. To this aim, the
development of methods complementary to the ground-based glaciological method is therefore
required. Since the 1970s, several methods have taken advantage of satellite imaging to
compute changes in glacier volume. Several glacier surface properties have thus been used as
proxies for volume fluctuations: changes in surface elevation from differencing digital
elevation models (DEM) (e.g., Berthier et al., 2016; Gardelle et al., 2013); end-of-summer
snow line elevation from high spatial resolution optical images (e.g. Meier and Post, 1962;
Rabatel et al., 2005, 2008, 2016); mean regional altitude of snow from low spatial resolution

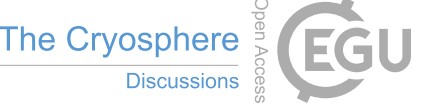

optical images (Drolon et al., 2016); or changes in the glacier surface albedo from high
temporal resolution images (Brun et al., 2015; Dumont et al., 2012; Sirguey et al., 2016).
Often used over icecaps or large ice masses, satellite derived DEM are not accurate enough to
compute confident annual volume changes of mountain glaciers, even if recent studies have
revealed promising results for multi-year glacier surface elevation changes of large
mountainous glacierized areas (Kääb et al., 2015). The method based on the correlation
between the regional snow cover and glacier SMB have shown satisfying results to retrieve
seasonal SMB, especially for the winter period. This method allowed the quantification of 55
glaciers SMB in the European Alps over the period 1998-2014 (Drolon et al., 2016). However,
this method still relies on calibration with field data and requires improvements for summer
and annual SMB. The method based on the identification on high spatial resolution optical
images of the end-of-summer snow line altitude has shown encouraging results in the French
Alps, multiplying by six the available long-term annual SMB time series (Rabatel et al., 2016),
but need to be automatized to compute glacier SMB at regional scales. In addition, monitoring
glacier surface properties on the daily or weekly basis and over large glacierized regions is still
challenging with high spatial resolution images. The current study is based on the albedo
method (Brun et al., 2015; Dumont et al., 2012; Sirguey et al., 2016). Images from the
MODerate resolution Imaging Spectroradiometer (MODIS) are processed to compute daily
albedo map of 30 glaciers in the French Alps over the period 2000-2015. Then, we rely on the
methodological framework proposed by Sirguey et al. (2016) on Brewster Glacier (New-
Zealand), looking at the relationships between annual and seasonal SMB and the glacier-wide
averaged surface albedo $\overline{\alpha}$. Our overall objective is to study the relationships between glacier
SMB and albedo by: (i) reconstructing the annual albedo cycle for 30 glaciers in the French
Alps for the period 2000-2015; (ii) linking the albedo signal to the seasonal components of the
SMB as well as to its annual values for 6 and 30 glaciers, respectively; (iii) assessing the
sensitivity of the retrieved albedo towards tuning parameters (cloud coverage threshold for



images processing, threshold on the winter albedo signal). Section 2 presents the available
SMB datasets used for the comparison and describes briefly the in situ automatic weather
stations (AWS) used to assess the quality of MODIS retrieved albedo. The method to retrieve
albedo maps is described in Sect. 3. Results are presented and discussed in Sect. 4 and 5. The
conclusion gathers the main results of the study and provides perspectives for future works.
**2    Study area and data**
**2.1    Site description**
The study focuses on 30 glaciers located in the French Alps (Fig. 1). Each glacier can be
classified as mountain glacier, extending over an altitudinal range from around 1600 m a.s.l.
(Argentière and Mer de Glace glaciers) to 4028 m a.s.l. (Blanc Glacier), and located between
the coordinates: 44°51" N to 46° N and 6°09" E to 7°08" E. The cumulative glacial coverage
considered in the present study is 136 km$^2$, i.e. half of the glacier surface area covered by 593
inventoried glaciers over the French Alps for the period 2006-2009 (Gardent et al., 2014).
Studied glaciers have been selected following four criteria related to the availability of field
data and remote sensing constraints, namely: (i) the annual glacier-wide SMB for the study
period had to be available; (ii) the glacier surface area had to be wide enough to allow robust
multi-pixel analysis; (iii) the glacier had to be predominantly free of debris to allow remotely-
sensed observations of the albedo of snow and ice surfaces; and (iv) seasonal SMB records
had to be available to consider seasonal variability. Finally, 11 glaciers have been selected in
the Ecrins range, 14 in Vanoise and 5 in Mont-Blanc (Fig. 1, and listed Table 1).





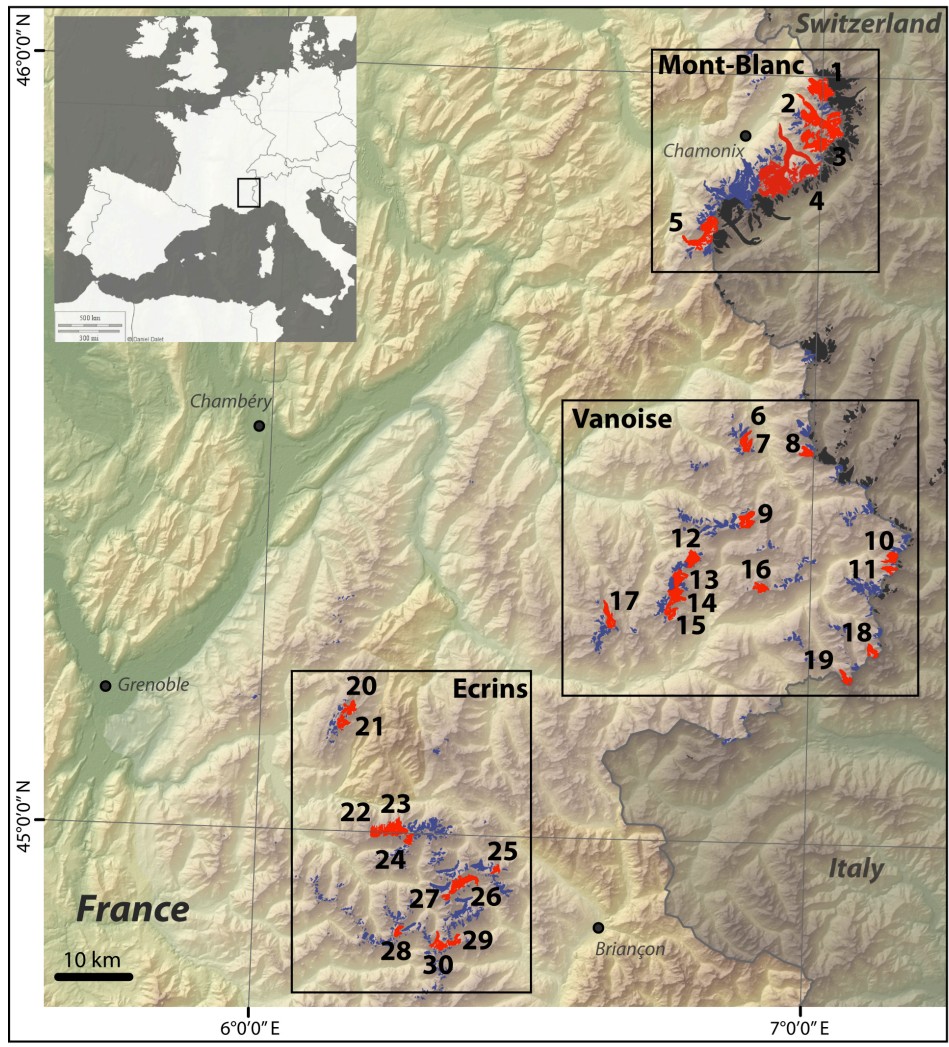

**Figure 1** Map of the region of interest with the studied glaciers shown in red (numbers refer to
Table 1). The four AWS used in the present study were set up on Saint-Sorlin Glacier (n°20).
Adapted from Rabatel et al. (2016).

## 2.2 MODIS satellite images

The MODIS sensor, onboard the TERRA - EOS/AM-1 satellite is acquiring near-daily images
of the Earth since February 25th, 2000. With 36 spectral bands ranging from 0.459 to 14.385
µm, and spatial resolution ranging from 0.25 to 1 km depending on the spectral band, MODIS



is nowadays one of the most used optical sensors for land surface observations. Because of its
short temporal revisit time, its long acquisition period and its moderate resolution, images
from MODIS are the most suitable for the present work. We therefore rely on about 15,000
MODIS calibrated Level 1B (L1B) swath images.

| # | Name | Mask size [Pixel] | $b_a = P_1^a \bar{\alpha}_a^{min} + P_2^a$ | | | | $b_s = P_1^s \bar{\alpha}_s^{int} + P_2^b$ | | | | $b_w = P_1^w \bar{\alpha}_w^{int} + P_2^w$ | | | |
|---|---|---|---|---|---|---|---|---|---|---|---|---|---|---|
| | | | $r^2$ | $RMSE$ | $P_1^a$ | $P_2^a$ | $r^2$ | $RMSE$ | $P_1^s$ | $P_2^s$ | $r^2$ | $RMSE$ | $P_1^w$ | $P_2^w$ |
| 1 | Tour | 71 | 0.78 | 0.61 | 14.9 | -7.8 | | | | | | | | |
| 2 | Argentière | 111 | 0.74 | 0.39 | 16.8 | -8.4 | 0.76 | 0.27 | 12.3 | -10.1 | 0.88 | 0.13 | 3.5 | -0.3 |
| 3 | Talèfre | 40 | 0.46 | 0.73 | 17.0 | -8.0 | 0.46 | 0.69 | 15.9 | -12.1 | 0.51 | 0.50 | 2.5 | -0.5 |
| 4 | Mer de Glace | 246 | 0.16 | 0.89 | 8.7 | -5.8 | 0.69 | 0.31 | 15.3 | -12.1 | 0.90 | 0.14 | 12.6 | -7.5 |
| 5 | Tré la Tête | 38 | 0.43 | 1.25 | 22.8 | -10.0 | | | | | | | | |
| 6 | Savinaz | 7 | 0.23 | 1.27 | 12.3 | -7.4 | | | | | | | | |
| 7 | Gurraz | 17 | 0.29 | 0.77 | 9.8 | -5.8 | | | | | | | | |
| 8 | Sassière | 19 | 0.52 | 0.67 | 8.2 | -4.9 | | | | | | | | |
| 9 | Grande Motte | 30 | 0.83 | 0.53 | 13.6 | -6.5 | | | | | | | | |
| 10 | Mulinet | 18 | 0.33 | 0.62 | 7.7 | -4.5 | | | | | | | | |
| 11 | Grand Méan | 11 | 0.44 | 0.64 | 7.8 | -4.2 | | | | | | | | |
| 12 | Arcelin | 37 | 0.64 | 0.52 | 6.6 | -3.7 | | | | | | | | |
| 13 | Pelve | 44 | 0.41 | 0.75 | 8.7 | -5.7 | | | | | | | | |
| 14 | Arpont | 41 | 0.28 | 1.0 | 9.8 | -5.8 | | | | | | | | |
| 15 | Mahure | 20 | 0.55 | 0.66 | 10.1 | -5.1 | | | | | | | | |
| 16 | Vallonnet | 19 | 0.36 | 0.66 | 3.4 | -2.0 | | | | | | | | |
| 17 | Gebroulaz | 23 | 0.62 | 0.45 | 9.1 | -4.6 | 0.76 | 0.28 | 9.8 | -7.9 | 0.36 | 0.19 | 1.6 | -0.1 |
| 18 | Baounet | 11 | 0.16 | 0.64 | 2.8 | -2.5 | | | | | | | | |
| 19 | Rochemelon | 11 | 0.31 | 0.67 | 4.3 | -2.8 | | | | | | | | |
| 20 | Saint-Sorlin | 31 | 0.86 | 0.37 | 13.8 | -6.3 | 0.94 | 0.21 | 14.7 | -11.0 | 0.75 | 0.19 | 2.3 | -0.5 |
| 21 | Quirlies | 15 | 0.60 | 0.54 | 11.4 | -5.2 | | | | | | | | |
| 22 | Mont de Lans | 35 | 0.69 | 0.64 | 11.4 | -5.4 | | | | | | | | |
| 23 | Girose | 60 | 0.70 | 0.43 | 9.1 | -4.7 | | | | | | | | |
| 24 | Selle | 13 | 0.79 | 0.41 | 9.0 | -4.4 | | | | | | | | |
| 25 | Casset | 7 | 0.73 | 0.47 | 8.9 | -4.6 | | | | | | | | |
| 26 | Blanc | 44 | 0.82 | 0.29 | 7.9 | -3.9 | 0.72 | 0.26 | 9.2 | -7.3 | 0.33 | 0.41 | 2.4 | -0.9 |
| 27 | Vallon Pilatte | 7 | 0.68 | 0.56 | 16.0 | -7.2 | | | | | | | | |
| 28 | Rouies | 14 | 0.72 | 0.68 | 18.0 | -7.8 | | | | | | | | |
| 29 | Sélé | 12 | 0.63 | 0.61 | 10.9 | -5.1 | | | | | | | | |
| 30 | Pilatte | 18 | 0.68 | 0.83 | 28.1 | -13.1 | | | | | | | | |


**Table 1:** List of studied glaciers, characteristics and albedo/mass balance correlations over
2000-2015, except for seasonal coefficients (over 2000-2010). For localization, refer to Fig. 1.
Highlighted rows exhibit glaciers where annual and seasonal in situ glacier-wide SMB data are
available. The mask size is expressed in number of pixels. To obtain the glacier mask area in
$km^2$, one should multiply the mask size by 0.0625 km². Determination coefficients are
expressed for each glacier (full plotted results are shown in supplementary material). Note the
units of r² (%), *RMSE, $P_1$ and $P_2$ (m w. e.).*

**2.3  Surface mass balance data**
In the French Alps, six glaciers allow both the seasonal and annual analyses to be conducted,
due to the availability of summer and winter SMB data ($b_s$ and $b_w$, respectively) obtained from



in situ measurements with the glaciological method (unpublished data, LGGE internal report,
listed Table 1). Among them, glacier-wide annual SMB $b_a$ of four glaciers (Argentière, Mer de
Glace, Gébroulaz and Saint-Sorlin glaciers) have also been calculated using the Lliboutry
approach (Lliboutry, 1974; Vincent, 2002; Vincent et al., 2000). The latter combines the
punctual in situ data and the glacier-wide surface elevation changes quantified from the
difference between DEM retrieved using aerial photogrammetry. In addition, glacier wide
annual SMB of the 30 studied glaciers were computed by Rabatel et al., 2016 using the end-
of-summer snow line measured on optical remote-sensing images and the glacier-wide mass
change quantified from DEMs differencing.
For the six glaciers where glacier-wide annual SMB are available from the two methods, i.e.,
in situ and satellite measurements, the average of the two estimates was used to calibrate and
evaluate the albedo method.
**2.4   In situ albedo measurements**
Albedo measurements acquired punctually using an AWS on Saint-Sorlin Glacier have been
used to evaluate the MODIS retrieved albedo. In situ albedo measurements were available for
three periods in the ablation zone (July-August 2006; June-August 2008; June-September
2009) and for one period in the accumulation zone (June-September 2008). Albedo data from
these AWS have been calculated as the ratio of the reflected to incident shortwave radiation
(0.3 to 2.8 μm) using two Kipp and Zonen pyranometers. With a potential tilt of the instrument
with respect to surface melting and the intrinsic sensor accuracy (±3%, Six et al., 2009), the
calculated albedo at the AWS shows a ±10% accuracy (Kipp and Zonen, 2009; Dumont et al.,

165   2012).

**3   Methods**
**3.1   MODImLab products**
MODIS L1B images were processed using the MODImLab toolbox (Sirguey, 2009). Image
fusion between MOD02QKM bands 1 and 2 at 250 m resolution and MOD02HKM bands 3 to
7 at 500 m resolution allows 7 spectral bands at 250 m resolution to be produced (Sirguey et





al., 2008). Then, atmospheric and topographic corrections are applied that include multiple
reflections due to steep surrounding topography (Sirguey, 2009). Various products are derived
from the corrected ground reflectance including snow and ice surface albedo (Dumont et al.,
2012). As recommended by Dumont et al. (2012) the WhiteSky (WS) albedo (estimated value
of the surface albedo under only diffuse illumination) is considered. The use of an anisotropic
reflection model for snow and ice has been preferred to the isotropic case, due to its closer
agreement with in situ measurements (Dumont et al., 2012). The MODImLab toolbox also
output sensor geometrical characteristics at the acquisition time such as the solar zenith angle
(SZA) and the observation zenith angle (OZA) used for post-processing the images (Sect. 3.4).
The MODImLab cloud detection algorithm is more conservative than the original MODIS
product (MOD35), and has been preferred as recommended in (Brun et al., 2015).
According to Dumont et al. (2012) and further assessed by (Sirguey et al., 2016) the overall
accuracy of MODImLab albedo product under clear-sky conditions is estimated at ±10%.
To mitigate the impact of shadows over the glaciers, MODImLab uses a DEM from the
Shuttle Radar Topography Mission (SRTM – 90 m resolution – acquired in 2000) to estimate
the sky obstruction by the surrounding topography and to correct the impact of shadows (see
Sirguey et al., 2009). The algorithm implemented in MODImLab is fully described in (Sirguey
et al., 2009) and inspired from (Dozier et al., 1981; Dozier and Frew, 1990) for the sky
obstruction factor processing (Horizon and Vsky in Sirguey et al., 2016), and from (Richter,
1998) for correction of shadows. It is first computed at 125 m resolution, providing Boolean-
type products of self and cast shadows per pixel. Results are then averaged and aggregated to
250 m resolution, producing sub-pixel fraction of shadow (further detailed in Sirguey et al.,
2009). Finally, MODIS data processed with MODImLab provides, among others, near-daily
maps of white-sky albedo at 250 m resolution together with cloud masks and cast and
projected shadows.
Albedo maps have been processed for 5,068 images for the Ecrins range, 4,973 for Mont-
Blanc and 5,082 for Vanoise over the period 2000-2015. Only images acquired between 09h50



and 11h10 AM UTC (+1h in winter and +2h in summer for local time conversion) were
selected to get minimum SZA and limit projected shadows of surrounding reliefs.
**3.2     Glacier masks**
Following Dumont et al. (2012) and Brun et al. (2015), we manually created raster masks of
the 30 glaciers, based on the glaciers' outlines from 1985-87 (Rabatel et al., 2013) and high
spatial resolution (6 m) SPOT-6 images from 2014. All debris-covered areas, together with
mixed pixel (rock-snow/ice) have been removed to capture only the snow/ice albedo signal.
The resulting number of pixels per glacier is listed in Table 1.
**3.3     Surface albedo and glacier-wide mass balance relationship**
**3.3.1     Basis of the method**
For glacier in the Alps (Dumont et al., 2012), the Himalayas (Brun et al., 2015) and in the
Southern Alps of New Zealand (Sirguey et al., 2016), the summer minimum glacier-wide
averaged albedo ($\overline{\alpha_a}^{min}$) has been significantly correlated to the glacier-wide annual SMB. This
relation allowed the glacier-wide annual SMB reconstruction from satellite images on the
Brewster Glacier, New Zealand, over the period 2000-2014 (Sirguey et al., 2016). The
relationship between $\overline{\alpha_a}^{min}$ and glacier-wide SMB results from the fact that solar radiation is the
main source of energy for melting snow and ice, both at the surface and within the first
centimeters below the surface (Van As, 2011). But this is not sufficient to explain why
averaged surface albedo is suitable for monitoring glacier SMB.
If we consider a temperate glacier in the mid-latitudes, its surface is fully covered by snow in
winter, leading to high and uniform surface albedo ($\overline{\alpha} \approx 0.8$ in Cuffey and Paterson, 2010).
During the ablation season, the accumulation area is still covered with snow conversely to the
ablation area where the ice is exposed and sometimes covered by debris. The overall albedo of
the glacier surface is therefore decreasing over the course of the ablation season, providing
information on the ratio of these two areas. The ratio between the size of the accumulation
zone and the entire glacier, called the accumulation-area ratio (AAR) has often been used as a



predictor of SMB both qualitatively (LaChapelle, 1962; Meier and Post, 1962; Mercer, 1961)
or quantitatively (Dyurgerov et al., 2009). Therefore, assessing $\overline{\alpha}_a^{\min}$ provides insights of the
relative share between the exposed ice and the snow-covered areas at the end of the ablation
season, also quantified by the AAR.
**3.3.2    From annual to seasonal surface mass balances**
In this study, $\overline{\alpha}_a^{\min}$ has been computed for the 30 glaciers in order to validate the method at a
regional scale. $\overline{\alpha}_a^{\min}$ occurs in summer, minimums out of summer are most likely artifacts.
Then, $\overline{\alpha}_a^{\min}$ has been directly correlated to available annual SMB data (listed in Table 1).
Following the work by Sirguey et al. (2016) on Brewster Glacier, a similar approach has been
used in order to validate the method at a seasonal scale but only on six glaciers (within our
sample of 30) for which the seasonal SMB are available. Conversely to Sirguey et al. (2016),
the summer SMB $b_s$ has been compared to the integrated albedo signal $\overline{\alpha}_s^{\mathrm{int}}$ during the entire
ablation season (1$^{st}$ May to 30$^{th}$ September) computed as follow and illustrated in Fig. 2.
$$\overline{\alpha}_s^{\mathrm{int}} = \int_{05.01}^{09.30} \overline{\alpha}(t).dt \qquad \text{Eq. (1)}$$


For the winter period (1$^{st}$ October to 30$^{th}$ April), the albedo signal has been computed similarly
than in Sirguey et al. (2016) by integrating the albedo when exceeding a threshold $\overline{\alpha}_T$,
considered as representative of fresh snowfall events (illustrated by the blue shaded area in
Fig. 2), as described by Eq. (2)):
$$\overline{\alpha}_w^{\mathrm{int}} = \int \overline{\alpha}(t) \begin{cases} \text{if } \overline{\alpha}(t) \text{ is found between 10.01 and 04.30} \\ \qquad \text{Only if } \overline{\alpha}(t) \geq \overline{\alpha}_T \end{cases}$$
$$\text{Eq. (2)}$$

The best threshold is the one maximizing the correlation between the retrieved cumulative
winter albedo $\overline{\alpha}_w^{\mathrm{int}}$ and the winter SMB. Threshold values have been computed independently
for each of the six seasonally monitored glaciers. To evaluate the impact of this threshold $\overline{\alpha}_w^{\mathrm{int}}$



has been computed without threshold over winter months (equivalent to $\overline{\alpha}_T = 0$). Finally,
hundred thresholds ranging from 0 to 1 every 0.01 have been tested to assess the sensitivity of
the method to $\overline{\alpha}_T$ (discussed Sect. 5). To compare each year together and remove the impact
of the variable integration time period for each glacier, both $\overline{\alpha}_s^{\,int}$ and $\overline{\alpha}_w^{\,int}$ have been divided by
the number of integrated days.

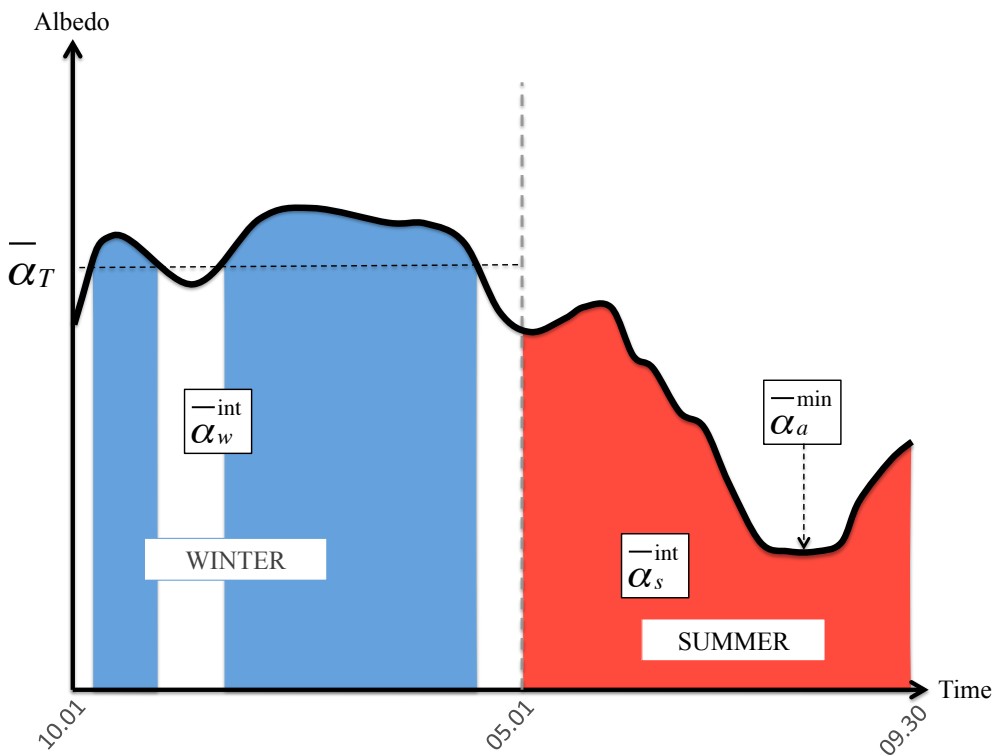


**Figure 2:** Schematic of a typical albedo cycle over one year, displaying parameters which
have been linked to annual, summer (between 1[st] May and 30[th] September in the northern
hemisphere) and winter (between 1[st] October and 30[rd] April in the northern hemisphere) SMB.
$\overline{\alpha}_w^{\,int}, \overline{\alpha}_s^{\,int}$ are retrieved using Eq. (2) and Eq. (1) respectively. $\overline{\alpha}_T$ represents an example of
threshold tested in Eq. (2). The summer minimum value of albedo is represented by $\overline{\alpha}_a^{\,min}$.



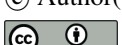

## 3.4 Data filtering

MODIS offers the opportunity to get daily images, but retrieving daily maps of Earth surface albedo remains challenging. Indeed, various sources of error require filtering the available images in order to only capture physical changes of the observed surface and not artifacts. Clouds are known to be a major problem in optical remote sensing of the Earth surface especially in the case of ice and snow covered surfaces. Even if some algorithms exist to differentiate clouds and snow-covered areas (e.g., Ackerman et al., 1998; Sirguey et al., 2009), omission errors are difficult to avoid, leading to erroneous albedo of the surface.

In this study, all images with a presence of cloud greater than 30% of the total glacier surface area have been discarded. This threshold is higher than that chosen in Brun et al. (2015) on the Chhota Shigri Glacier (20%), and we thus discuss Sect. 5.1 the impact of the computed cloud threshold on the derived albedo results. When determining $\overline{\alpha}_a^{min}$, 0% of cloud cover has been imposed as a condition and visual check for each year and each glacier has been performed. Snapshots from the fusion of MODIS bands 1 to 3 and from bands 4 to 6 (Sirguey et al., 2009) have been used to visually check the images, together with images from other satellites (mostly from the Landsat archive) and pictures and comments from mountaineering forums. This last step, although laborious when studying 30 glaciers allowed the identification of the summer minimum to be improved. Visual check of the images also confirms that projected shadows of clouds are not affecting the albedo map. Another source of error is the impact of the OZA. As mentioned in Sirguey et al. (2016), accuracy of the MODIS retrieved albedo strongly decreases for viewing angles above 45° as pixel size increases from 2 to 5-folds from OZA = 45° to 66° (Wolfe et al., 1998). This phenomenon is accentuated when observing steep-sided snow/ice surfaces, surrounded by contrasted surfaces (rocks, forests, lakes...). This distortion could lead to capture the mean albedo of a glacier plus its surroundings. Following this, we decided to filter the images according to their OZA angle, as further described Sect. 4.1.




## 4   Results

### 4.1   Retrieved albedo assessment

A quantitative evaluation of the retrieved albedo has been performed with AWS deployed on Saint-Sorlin Glacier. Measurements have been synchronized between punctual albedo for MODIS and a 2-hour averaged albedo around MODIS acquisition time for the AWS. It is worth reminding some differences between the in situ measured albedo data and the one retrieved using MODIS. The downward facing pyranometer stands at around 1 m above the surface, corresponding to a monitored footprint of *ca.* 300 m$^2$ (theoretical value for a flat terrain) while the pixel area of MODImLab products matches 62,500 m$^2$. Quantified albedos from each method are therefore not representative of the same area. On the other hand, incoming radiation data are extremely sensitive to a tilt of the sensor located on the AWS and maintaining a constant angle throughout the monitoring period remains challenging, especially during the ablation season. For instance, a tilt of 5° of the pyranometer at the summer solstice can increase by 5% the error on the irradiance measurement (Bogren et al., 2016). No sensor tilt was deployed on the AWS, thus preventing the application of tilt-correction methods (e.g., Wang et al., 2016). Nonetheless, regular visit allowed to maintain the sensor horizontal and to limit errors in the irradiance measurements.





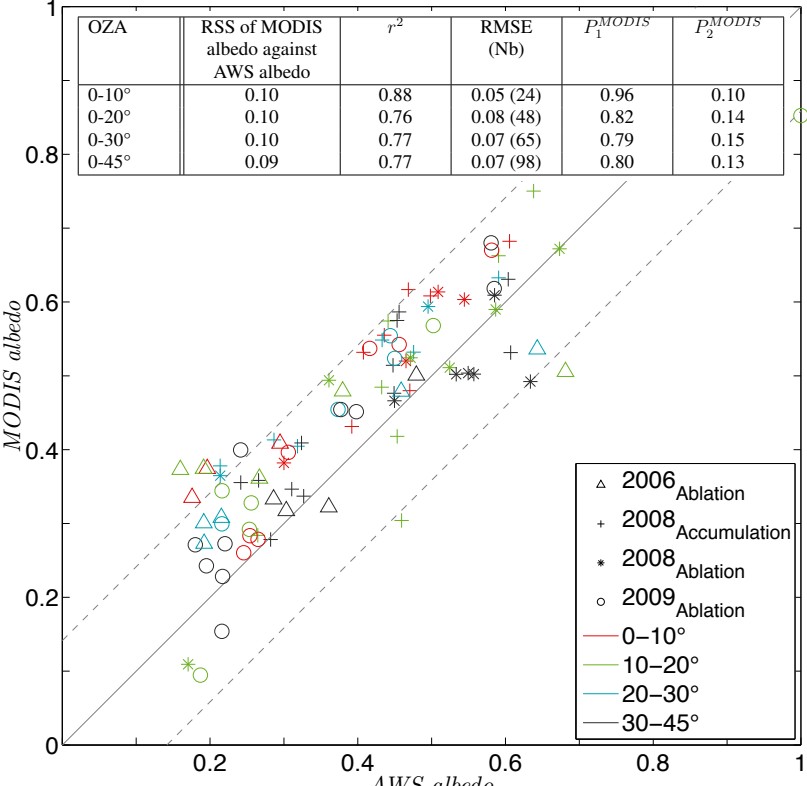

**Figure 3:** MODIS albedo and AWS albedo data for different OZA classes on Saint-Sorlin

Glacier. Years indicated in the caption correspond to the year of acquisition while subscripts

express the AWS location in the accumulation or ablation areas. The mean discrepancy

between MODIS and AWS albedo per OZA is quantified by the RSS (residual sum of square).

Correlation coefficient per OZA classes are also provided, with $r^2$, RMSE together with the

number of compared measurements (Nb), and coefficients of the equation:

$MODIS_{albedo} = P_1^{MODIS} AWS_{albedo} + P_2^{MODIS}$ . The continuous grey line illustrates the 1:1

relationship between AWS and MODIS retrieved albedo. Thin and dotted lines represent the

combined uncertainties on both AWS and MODIS retrieved albedo (absolute value of 10% for

each), only accounting for intrinsic sensor accuracy and not for errors related to the acquisition

context, e.g. size of the footprint.





Figure 3 illustrates the comparison between the retrieved and measured albedos at the AWS
locations for various OZA classes. One can note minor differences between the data plotted in
Fig. 3 and those presented in Dumont et al. (2012, Fig. 2). These differences are related to
changes in the MODImLab algorithm and different computation of the in situ albedo,
integrated over a two-hour period in the current study.
In Fig. 3, the spread between MODIS and AWS albedos is higher for low albedos (i.e. ablation
area). This is related to the footprint difference as described earlier, accentuating the albedo
differences when monitoring heterogeneous surface (snow patches, melt pounds…), even
more pronounced in summer. One can also note that MODIS albedo often over-estimate the
AWS albedo value. This over-estimation could be explained by: (1) the MODImLab albedo
retrieval algorithm. Indeed, under-estimation of the incoming radiation computed in the
MODImLab algorithm would lead to over-estimated retrieved albedo values, in addition the
atmospheric corrections used to compute the incident radiation could be hypothesized as
source of error (*e.g.* modeled transmittance through a simplified computed atmosphere, refer
to (Sirguey et al., 2009) for further description); (2) the AWS albedo measurements. Indeed,
view angles of AWS pyranometers (170°) could influence the retrieved albedo by monitoring
out-of-glacier features (*e.g.* moraines, rock walls, ...), resulting in under-estimated albedo
values. However, it is worth noting that most of the points are within the combined uncertainty
of both sensors and these differences in albedo retrieved from MODIS and the AWS are thus
hard to interpret.
Finally, Fig. 3 shows substantial differences between OZA<10° and other OZA classes. For
OZA<10, MODIS albedos better agree with AWS albedos than for the three other classes.
Integrating MODIS images with OZA>10° substantially deteriorate the agreement with AWS
albedos (in term of $r^2$, *RMSE* and the slope $P_1^{MODIS}$), especially on "narrow" targets as alpine
mountain glaciers. We therefore chose to prioritize images acquired with low OZA to avoid
detection of non-glacierized surfaces. Therefore, four classes of images have been selected
following the criteria presented in Table 2.



| Class | OZA (°) | Criteria |
|---|---|---|
| I | $OZA \leq 10$ | All retained |
| II | $10 < OZA \leq 20$ | Retained if more than 7 days between consecutive images from class I |
| III | $20 < OZA \leq 30$ | Retained if more than 7 days between consecutive images from class I+II |
| IV | $OZA > 30$ | Not retained |

**Table 2**: Filtering the images from OZA values.

For the rest of the computation, the absolute ±10% accuracy per pixel estimated in Dumont et al. (2012) has been considered. We determined the uncertainty on $\bar{\alpha}$ by accounting for the spatial variability of the albedo signal within the glacier and considering that our sets of pixels are independent from each other (Eq. (3)):

$$\sigma_{\bar{\alpha}} = \frac{\sigma}{\sqrt{N}} \qquad \text{Eq. (3)}$$

where $\sigma$ stands for the standard deviation of the pixels albedo with $N$ the number of pixels.

**4.2 Temporal variability of the albedo signal**

Using the "step-by-step" filtering procedure explained in Sect. 3.4, the ~16-yr albedo cycle of each of the 30 glaciers was obtained (results available in the supplementary material). Figure 4 illustrates the entire albedo time series for Saint-Sorlin Glacier over the period 2000-2015. We observed that the albedo decreases from the beginning of summer (dashed red line), reaching $\bar{\alpha}_a^{min}$ in August/September and rising again at the end of September. This cyclicality is a proxy of surface processes. The snow cover decreases at the beginning of summer until reaching its lowest extent, and finally increases again with the first snowfall in late summer to reach its maximum extent in winter/spring.




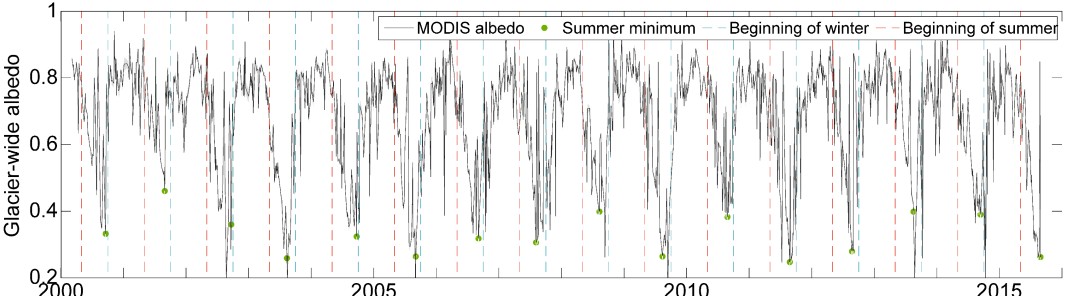

**Figure 4:** ~16-yr albedo course for Saint-Sorlin Glacier. Glacier-wide averaged albedo is represented with the continuous black line. The green dots spot for each summer the minimum average albedo, and have been manually checked for all years and glaciers. Dashed red and blue lines stand for the beginning of the defined ablation and accumulation seasons (May and October 1[st] respectively).

The periodicity of the albedo signal is however not so well defined for some of the studied glaciers. For instance, Argentière Glacier exhibits a severe drop of $\overline{\alpha}$ in winter, reaching values as low as summer minimums ($\overline{\alpha} \approx 0.4$). The observed drop of albedo in winter occurs during more than one month centered on the winter solstice (December 21[st]) and is observed for nine glaciers (Argentière, Baounet, Casset, Blanc, Girose, Pilatte, Vallon Pilatte, Tour and Sélé glaciers, refers to supplementary material for full results). These glaciers are located within the three studied mountain ranges but have the common characteristic to be very incised with steep and high surrounding faces. We studied the albedo series as a function of the SZA to reveal possible shadowing on the observed surfaces. Figure 5 displays the same cycle as Fig. 4 for Argentière Glacier but providing information about SZA. As a reminder, the MODImLab white-sky albedo is independent of the illumination geometry but the computed albedo for each pixel can be subject to shadowing from the surrounding topography. Two mains observations stands out from the winter part of the cycle in Fig. 5: (i) most of MODIS $\overline{\alpha}$ severely decrease under $\overline{\alpha} = 0.6$ for SZA greater than 60° corresponding to



383 November to January images, (ii) these drops are not systematic and we rather observe a

384 dispersion cone than a well-defined bias. As there are no physical meanings to systematic

385 change of the surface albedo during a part of the winter period and owing to the fact that this

386 dispersion is only observed for topographically incised glaciers, these decreases in albedo have

387 been considered as artifacts. These observations led us to carefully process winter albedos and

388 to perform a sensitivity study on the impact of the threshold albedo parameter $\overline{\alpha}_T$ .

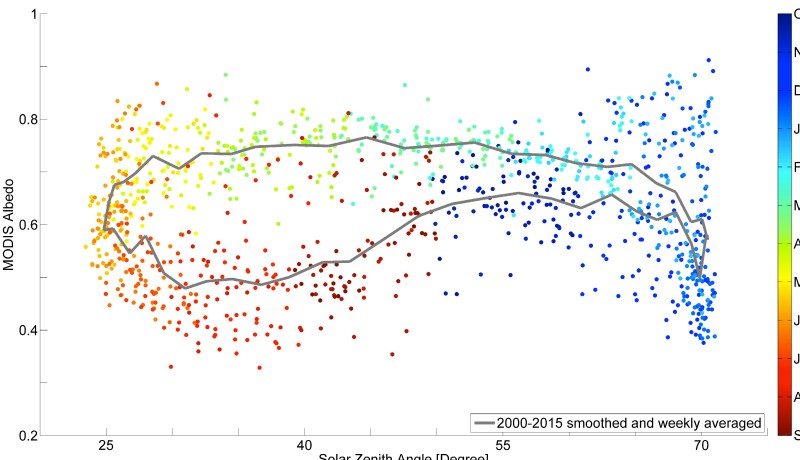

389

390 **Figure 5:** Albedo cycle for Argentière Glacier as a function of the SZA. Each point

391 corresponds to glacier-wide averaged albedo for each available image. The 16 years are

392 displayed. Color scale gives indication on the date of the used image. The thick grey line

393 describes the weekly albedo averaged over the entire study period. For readability purpose, the

394 averaged albedo has been smoothed, using a 7 points running average.

395

396 **4.3 Albedo and glacier-wide surface mass balance**

397 **4.3.1 $\overline{\alpha}_a^{\min}$ and annual surface mass balance**

398 The summer minimum average albedo for each year and each glacier has been linearly

399 correlated to the glacier-wide annual SMB. Figure 6 illustrates the relationship between $\overline{\alpha}_a^{\min}$





and $b_a$ for Blanc Glacier (all the other glaciers are shown in the supplementary material). Error
bars result from the dispersion of the SMB dataset for each year, and from the glacier intrinsic
variability of the albedo signal the day of $\overline{\alpha}_a^{\min}$ acquisition. For the glaciers where the glacier-
wide annual SMB is available from the SLA method, the uncertainty is about ±22 *cm w.e.* on
average (ranging from 19 to 40 *cm w.e.* depending on the glacier, Rabatel et al., 2016).

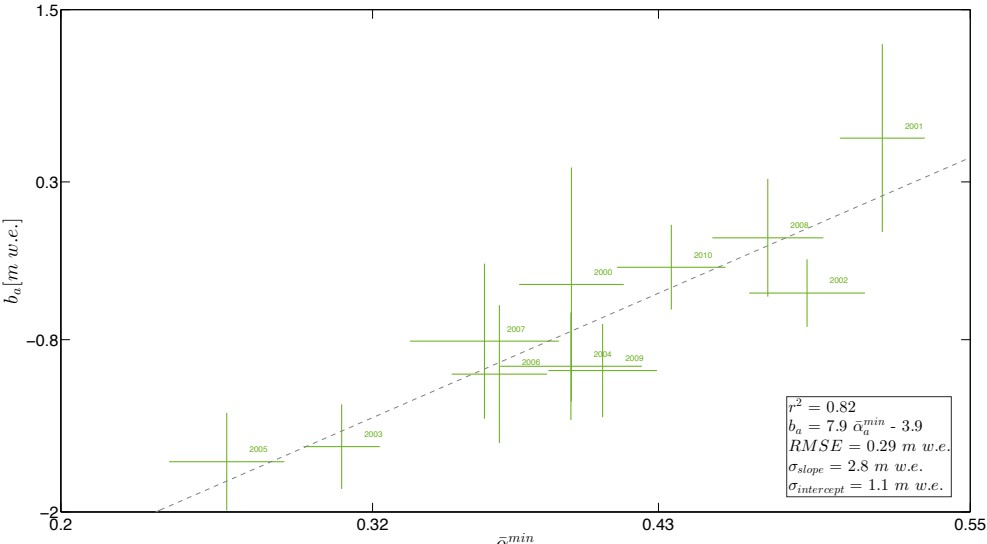


**Figure 6:** Annual SMB as a function of the MODIS retrieved summer minimum glacier-wide
average albedo for Blanc Glacier. Error bars result on the dispersion of the available annual
SMB data and on the quadratic sum of the systematic errors made on each albedo
measurement. The thin dashed grey line illustrates the line of best fit, along with regression
coefficients and significance.

Twenty-seven glaciers show significant correlations (refer to Table 1 for full results) if
considering a risk of error of 5% (according to a Student's t test). However, the linear
correlation has no statistical significance for three glaciers with $r^2 < 0.25$. A possible
explanation is the high number of removed images in summer due to manually checked thin
overlying clouds not detected by the MODImLab cloud algorithm.



Looking at the 27 glaciers for which significant relationships have been found, 2001 is
regularly identified as an outlier. According to existing SMB datasets, 2001 is the only year of
the period 2000-2015 for which the annual SMB has been positive for all the studied glaciers
(0.80 m w.e. yr$^{-1}$ in average). Hence, according to computed determination coefficients in
Table 1, correctly predicting the surface mass balance values for the year 2001 using the
albedo method would imply to monitor a high value of minimum glacier-wide average albedo,
often greater than 0.7 (i.e. 0.83 and 0.95 for Rochemelon and Vallonnet glaciers, respectively).
Taking into consideration snow metamorphisms during the summer period, melting at the
surface and possible deposition of debris or dusts, monitoring such high albedo values
averaged at the glacier scale is unrealistic. Furthermore, removing 2001 from the time series
does not increase the number of glaciers for which the correlation is significant.
Finally, this study confirms the robust correlation between $\overline{\alpha_a}^{min}$ and $b_a$ for 27 of the 30 studied
glaciers. It also reveals some limitations by under-estimating the annual SMB value for years
with very positive annual SMB.
**4.3.2  $\overline{\alpha_s}^{int}$ and summer surface mass balance**
Studying the integral of the albedo signal during the ablation season can provide insights on
the intensity of the ablation season and thus on the summer SMB $b_s$. As described in Sect.
3.3.2, $\overline{\alpha_s}^{int}$ has been computed and connected to the in situ $b_s$. Figure 7 illustrates the results
for Saint-Sorlin Glacier.

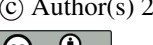



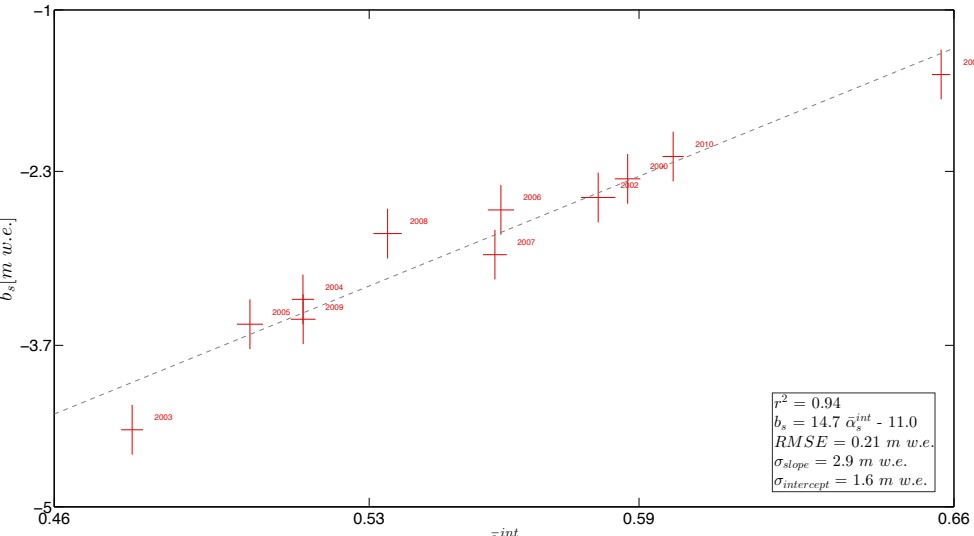


**Figure 7:** Summer SMB $b_s$ expressed as a function of the integrated albedo over the entire

ablation season for Saint-Sorlin Glacier. Error bars result from the uncertainties related to the

glaciological method (measurements and interpolation at the glacier scale of the punctual

measurements, ±20 *cm w.e.* in total), and on the quadratic sum of the systematic errors made

on each albedo measurement. Thin dashed grey line represents the linear regression showing

the best correlation between the two variables, together with correlation coefficients.

443

Saint-Sorlin Glacier, together with the five other seasonally surveyed glaciers showed a

significant correlation between the two observed variables (from $r^2 = 0.46$ to $r^2 = 0.94$ with an

error risk < 5%, all statistics referred in Table 1). Conversely to $\overline{\alpha}_a^{min}$, $\overline{\alpha}_s^{int}$ is slightly more

robust to the presence of undetected clouds as its value does not rely on a single image. The

lowest correlation has been found for Talèfre Glacier. The latter accounts for a relatively large

debris-covered tongue that has been excluded when delineating the glacier mask (see

supplementary material). Consequently, the low correlation could be partly explained by this

missing area, considered in the glaciological method but not remotely sensed. To conclude,



$\overline{\alpha}_s^{int}$ has been significantly correlated to $b_s$ and is therefore a reliable proxy to record the
ablation season.

### 4.3.3   Retrieval of winter surface mass balance

As described Sect. 3.3.2, a similar method to the one used by Sirguey et al. (2016) has been
applied for winter SMB quantification. Figure 8 illustrates the computed correlation for
Argentière Glacier. Results for the five other seasonally investigated glaciers are listed in
Table 3. The use of a glacier-dependent threshold value $\overline{\alpha}_T$ substantially improves the
correlation between winter SMB and $\overline{\alpha}_w^{int}$ for three of the six glaciers (Saint-Sorlin, Blanc and
Talèfre glaciers).

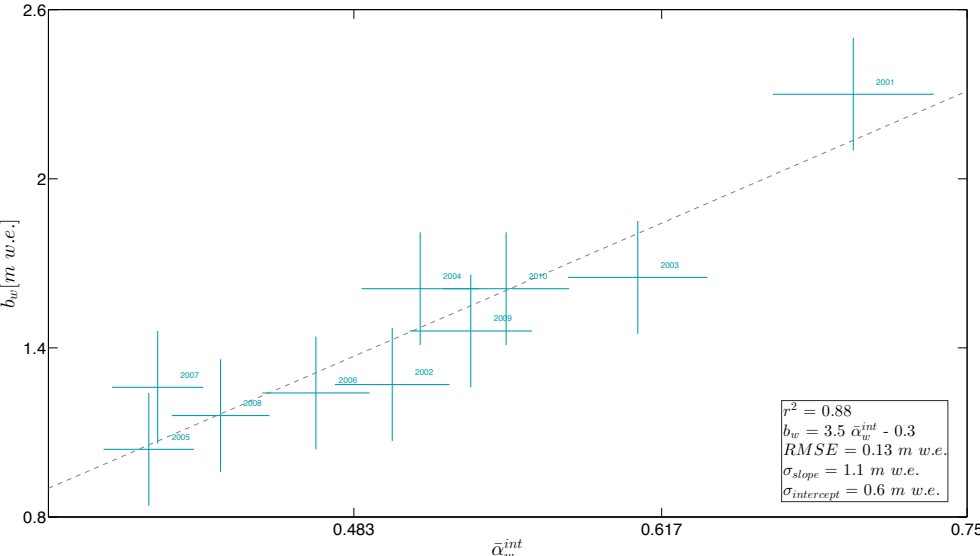


**Figure 8:** Winter SMB, $b_w$, expressed as a function of the integrated albedo over the entire
accumulation season for Argentière Glacier. Winter SMB of 2001 corresponds to the winter
2000/2001. Error bars result from the uncertainties related to the glaciological method and on
the quadratic sum of the systematic errors made on each albedo measurement. Thin dashed
grey line represents the linear regression showing the best correlation between the two
variables, together with correlation coefficients.





For Argentière, Mer de Glace and Gébroulaz glaciers, a significant correlation is found
whatever the value of the albedo threshold $\bar{\alpha}_T$ is (Table 3). Furthermore, $\bar{\alpha}_T$ is far from being
uniform on the six glaciers ($0.53 \geq \bar{\alpha}_T \geq 0.76$). We therefore reconsider the idea of using a
threshold as a representative value of fresh snowfall, as there is no physical reason that this
threshold varies, at least within the same region.

| Glacier | $\bar{\alpha}_T$ | $r^2$ using $\bar{\alpha}_T$ | $r^2$ without $\bar{\alpha}_T$ |
|---|---|---|---|
| Saint-Sorlin | 0.76 | 0.75 | 0.21 |
| Argentière | 0.58 | 0.88 | 0.76 |
| Talèfre | 0.68 | 0.59 | 0.25 |
| Mer de Glace | 0.53 | 0.90 | 0.87 |
| Gebroulaz | 0.75 | 0.36 | 0.25 |
| Blanc | 0.70 | 0.33 | 0.21 |


**Table 3:** Coefficients of determination for the relationship between the winter SMB $b_w$ and the
integrated winter albedo, computed with and without the albedo threshold $\bar{\alpha}_T$.

**5 Discussion**
In this section, we first discuss the impact of the threshold applied to the cloud cover fraction
on the obtained results. Then, the observed discrepancies and artifacts of the winter albedo
signal on some of the studied glaciers have been analyzed through a sensitivity study focused
on the algorithm correcting the shadows. Afterward, we discuss the sensitivity of the
correlation, between $\bar{\alpha}_w^{int}$ and $b_w$, toward the selected albedo threshold $\bar{\alpha}_T$. We finally express
the main limitations and assessments of the albedo method.
**5.1 Cloud coverage threshold**
As stated in Sect. 3.4, a value of 30% of cloud coverage over the glacier mask has been
defined as the acceptable maximum value for considering the albedo map of the day. We
computed a sensitivity study on the impact of this threshold on the value of the obtained
correlations between the integrated summer albedo and the in situ summer SMB. The summer
period has been chosen as it represents the period when the albedo of the glacier is the most



contrasted, between bare ice and snow/firn. The glacier-wide average albedo in this period is
therefore more sensitive to possible shading of a part of the glacier. Figure 9 illustrates the
results for the six seasonally surveyed glaciers. The used value of the allowed cloud coverage
appears not to have a substantial impact on the correlation. This observation first implies that
the MODImLab cloud product is reliable enough to only compute surface albedo and to avoid
too frequent misclassification between the clouds and the surface. It also suggests that
removing too many images because of partial cloud cover removes information about the
glacier-wide average albedo variability. However, allowing all images, even when the glacier-
wide average albedo is computed on only 10% of the glacier (90% of detected cloud
coverage), does not reduce significantly the correlation for most of the six glaciers.

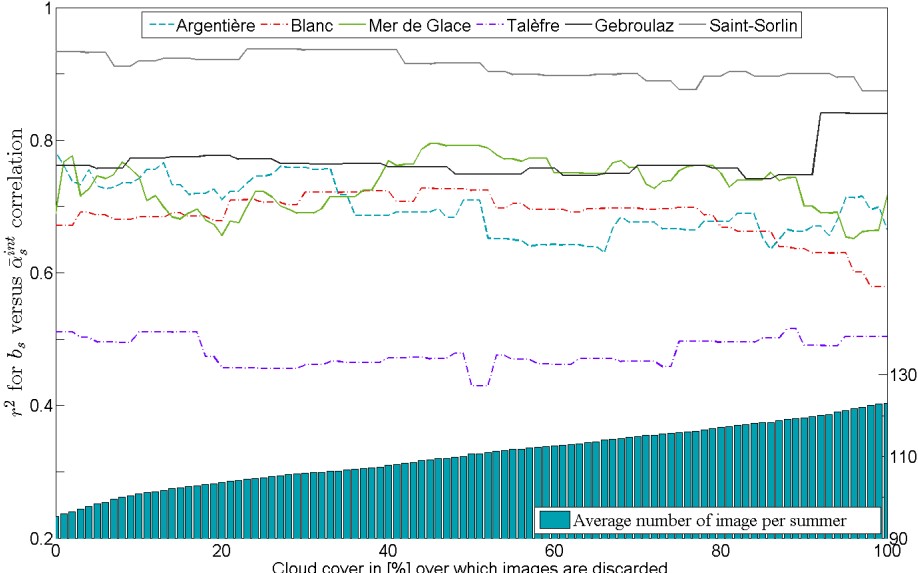


**Figure 9:** $r^2$ for the six seasonally surveyed glaciers for the albedo summer integral *versus*
summer SMB relationship against the cloud threshold above which images have been
discarded during the summer season. For the computation, hundred thresholds have been
tested between 0 and 100%. The inner histogram illustrates the number of considered images
per summer and averaged on the six glaciers.



Nevertheless, hypothesizing that the glacier-wide average albedo of a small fraction of the
glacier (e.g., 10%) is suitable to represent the entire glacierized surface is questionable. It
therefore depends on the size of the observed glacier, where 10% of a glacier of 3 and 30 km$^2$
have not the same meaning, but also on the delineated mask (ablation area not entirely
considered because of debris coverage...). The summer-integrated albedo is also highly
dependent on the time gap between useful images. In other words, if an image has an
"anomalous" glacier-wide average albedo because of high cloud coverage, the impact on the
integrated value will be smaller if "normal condition" albedos are monitored at nearby dates.
The average number of available images per year does not largely differ between the various
computed cloud coverage thresholds. It varies in average from 95 to 123 images per summer
period for respectively 0% and 100% cloud coverage threshold. Intermediate values are 106,
111 and 116 images per summer for 30, 50 and 75% cloud coverage threshold, respectively.
The difference in significance of $r^2$ (according to a Student's $t$ test) between opting for 0% and
100% is almost negligible, and choosing the best cloud threshold value is rather a compromise
between the number of used images and the resulting correlation with glacier-wide SMB. We
finally concluded that selecting cloud coverage threshold to 30% presents the best
determination coefficients between the integrated summer albedo and the summer balance for
most of the six glaciers without losing too much temporal resolution.
**5.2    Evaluation of winter albedo values**
In light of the documented dispersion on $\bar{\alpha}$ during some of the winter months on several
studied glaciers (Sect. 4.2), sensitivity of the MODIS retrieved albedo against correction of
shadows had been assessed. This work has only been conducted on the 250 m resolution raster
products and specifically on the cast shadow product because self-shadow corrections can be
considered as reliable enough because only related to the DEM accuracy. We thus defined a
pixel as "corrected" when at least one of its sub-pixels was classified as shadowed. From then
on, two glacier-wide albedos $\bar{\alpha}$ have been defined: (i) $\bar{\alpha}_{\text{non-cor}}$ computed on non-corrected



pixels only, classified as non-shadowed; (ii) $\bar{\alpha}$ of both corrected and non-corrected pixels,
equal to the glacier-wide average albedo. Figure 10 illustrates the difference between $\bar{\alpha}_{\text{non-cor}}$
and $\bar{\alpha}$ as a function of the percentage of corrected pixels over the entire glacier. The study
was performed on Argentière Glacier (111 pixels) that exhibited large $\bar{\alpha}$ artifacts in winter
(Fig. 5). The inner diagram allows emphasizing the annual "cycle" of modeled shadows,
contrasted between nearly no cast shadows in summer and an almost fully shadowed surface
in winter. We represent the 1 standard deviation of $\bar{\alpha}$, averaged by classes of 5% corrected
pixels. In other words, it illustrates the mean variability of the glacier-wide surface albedo.
Therefore, for images with $\bar{\alpha}_{\text{non-cor}}$ - $\bar{\alpha}$ within the interval defined by 1 st.dev. of $\bar{\alpha}$, errors
resulting from the correction algorithm are smaller than the spatial variability of the glacier-
wide albedo glacier. We also selected only significant values, following a normal distribution
of the averaged $\bar{\alpha}$. Consequently, only values at $\pm 1\sigma$ (68.2%) in term of percentage of
corrected pixel have been retained (i.e. when the relative share of corrected pixels ranged from
15.9 to 84.1%). Between 0 and 15.9%, $\bar{\alpha}_{\text{non-cor}}$ and $\bar{\alpha}$ are not sufficiently independent
because of low number of corrected pixels, and beyond 84.1%, $\bar{\alpha}_{\text{non-cor}}$ is computed over a too
small number of pixels. As a consequence, even if the albedo correction in the shadowed parts
of the glacier could be improved, most of the errors related to this correction do not depreciate
the results. Above 80% of corrected pixels (December to early February), differences between
$\bar{\alpha}_{\text{non-cor}}$ and $\bar{\alpha}$ exceed the monitored spatial variability of $\bar{\alpha}$. These anomalies are at the root
of the observed artifacts Fig. 5 by the severe drops of albedos and described Sect. 4.2.





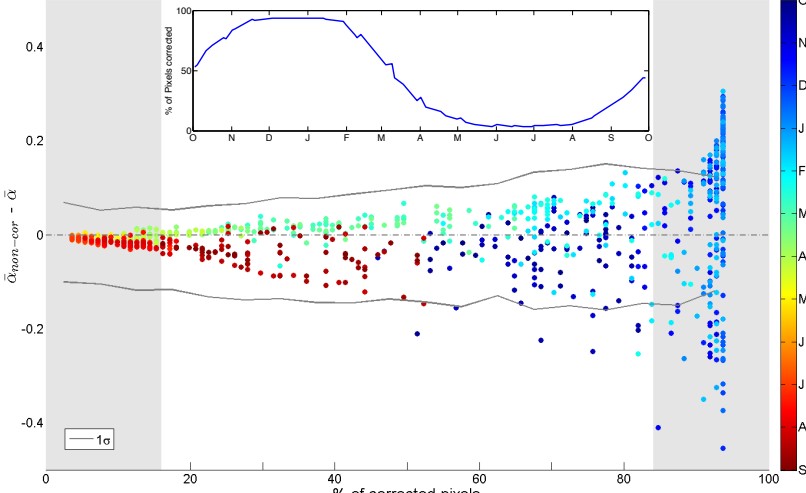


**Figure 10:** Impact of the ratio of corrected pixels toward the difference between non-corrected

and glacier-wide albedo. Each point corresponds to one acquisition and the 16 years are

therefore displayed on this graph. Color scale gives indication on the date of the acquired

image. Grey shaded areas correspond to ratios of corrected pixels for which $\overline{\alpha}_{\text{non-cor}}$ - $\overline{\alpha}$ has

low statistical robustness (refer to the main text). Thin grey lines represent $1\,\sigma$ standard

deviation of $\overline{\alpha}$, averaged by classes of 5% corrected pixels. The inner graph illustrates the

amount of corrected pixel, function of the selected month.

562

In addition, a seasonality in the albedo signal can be observed with $\overline{\alpha}_{\text{non-cor}}$ - $\overline{\alpha}$ > 0 in early

spring (February to April) while $\overline{\alpha}_{\text{non-cor}}$ - $\overline{\alpha}$ < 0 in summer and autumn (June to November).

This could be explained by different localizations of shadowed area for a given ratio of

corrected pixel. As an example, a glacier could have in October a snow- and shadow-free

snout and a covered by fresh snow and shadowed upper section. This configuration would

induce a negative difference as we observe from June to November. Conversely, this glacier

could present in March (same ratio of corrected pixels than October) a complete snow





coverage, leading to a smaller difference between $\overline{\alpha}_{\text{non-cor}}$ and $\overline{\alpha}$ (<0.1) that could even result
on positive difference as we observe from February to April.
Finally, observed albedo artifacts in winter are most likely due to the correction of shadows.
On the other hand, correcting shadows accurately and consistently is extremely challenging.
As illustrated by Fig. 10, a way to confidently consider the albedo signal is to exclude values
with too large share of corrected pixels. However, because of the inter-annual approach carried
out in this study, such systematic artifact is not depreciating the results but would be a major
issue on studies focused on albedo values themselves (e.g. maps of snow extents…).
**5.3    Evaluation of the winter albedo threshold**
The albedo threshold, $\overline{\alpha}_T$, for which the winter albedo signal is integrated is considered in
Sirguey et al. (2016) as representative of the presence of fresh snow at the glacier surface. In
order to study the impact of $\overline{\alpha}_T$ on the correlation between the winter integrated albedo and
the in situ winter SMB, we computed the $r^2$ considering 100 values of $\overline{\alpha}_T$ (from 0 to 1).
Figure 11 displays the computed results for the six seasonally surveyed glaciers. No glacier
provides the same threshold maximizing $r^2$. For Argentière and Mer de Glace, using a
threshold does not drastically maximize the relation and the integral can be processed without
using a threshold. These two glaciers also provide the best correlation coefficients compared
to the other four glaciers and are by far the largest glaciers of our monitoring set (14.59 and
23.45 km$^2$ for Argentière and Mer de Glace glaciers respectively). Indeed, a possible
explanation of this good correlation, even without threshold, relies on the morpho-topographic
features on these two large glaciers. With a glacier snout reaching 1600 m a.s.l., the tongue of
these glaciers can experience melting events (resulting in contrasted pixels in terms of albedo
value), even during the winter season. The glacier-wide albedo therefore provides a good
proxy of the winter SMB on the glacier because of the large altitudinal range of the glacier.
For Saint-Sorlin Glacier, a threshold of 0.76 improves significantly the correlation, similarly to
the threshold found for Brewster Glacier by Sirguey et al. (2016). We can mention the analogy

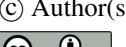



between Saint-Sorlin and Brewster, having similar morpho-topographic features, in terms of
surface area, general aspect and slope. Talèfre Glacier, with $\overline{\alpha}_T$ equals 0.68, is the second
glacier for which using a threshold significantly improves the correlation.

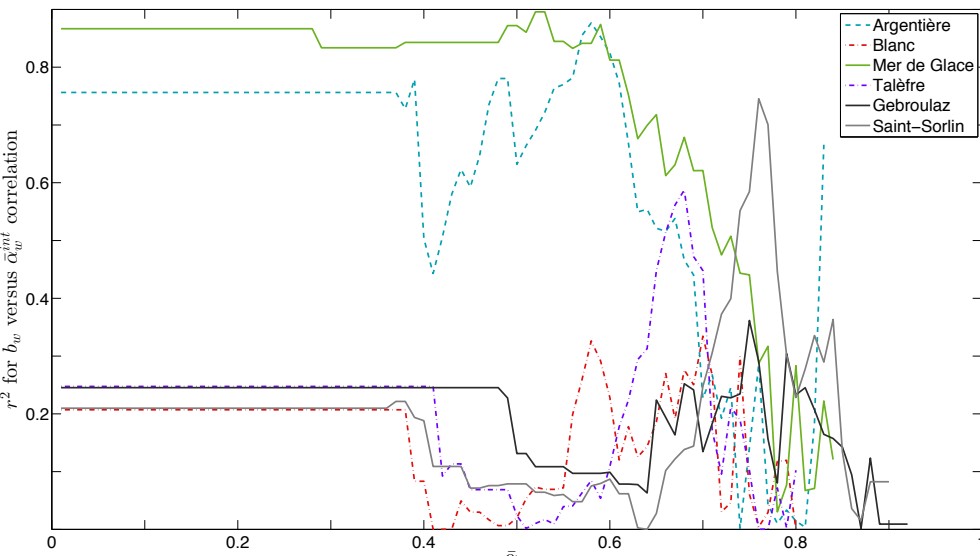


**Figure 11:** $r^2$ of the albedo winter integral *versus* winter SMB relationship as a function of $\overline{\alpha}_T$
chosen for the winter integration for the six seasonally monitored glaciers. Hundred thresholds
have been tested between 0 and 1.

Nevertheless, all studied glaciers, apart from Mer de Glace, present a low robustness against
$\overline{\alpha}_T$ and important and non-physical variations of $r^2$ occur for most of the computed $\overline{\alpha}_T$. The
method is preferentially filtering years with repeated low albedos. Therefore, for a given
threshold $\overline{\alpha}_T$, some years are necessarily more filtered than others resulting in large
degradation or enhancement of the $b_w$ *vs.* $\overline{\alpha}_w^{int}$ correlation. As an example for Argentière
Glacier (Fig. 8 and Fig. 12 with $\overline{\alpha}_T$ equals 0.58 and 0.42 respectively), opting for $\overline{\alpha}_T$ =0.42 is
affecting preferentially years 2001 and 2004 and therefore reduces significantly the

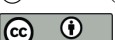


correlation. On the other hand, given threshold can by chance, favors the $b_w$ vs. $\overline{\alpha}_w^{\text{int}}$
correlation.

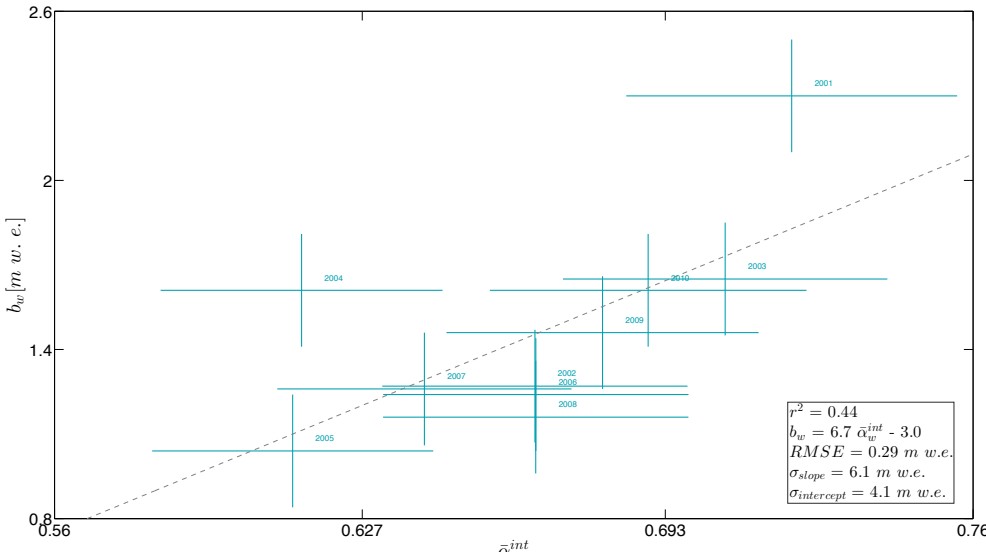

**Figure 12:** Similar relationship between $b_w$ vs. $\overline{\alpha}_w^{\text{int}}$ but for $\overline{\alpha}_T$ =0.42. Axis, legend and
uncertainties are identical to Fig.8.

These results finally question the use of $\overline{\alpha}_T$ as a threshold detecting only fresh snowfall events
and seems to maximize artificially the correlation between $b_w$ and $\overline{\alpha}_w^{\text{int}}$. Large glaciers (>10
km²) appear to be more robust for threshold-free detection but further studies on a more
exhaustive set of large glaciers would be required.
**5.4    Limits of the albedo method**
In agreement with Dumont et al. (2012) and Brun et al. (2015), retrieving the glacier annual
SMB from albedo summer minimums proves to be an efficient method. Low correlations often
result from high and persistent cloud coverage during summer, reducing the chance of spotting
the albedo summer minimum. For SMB reconstruction purpose, a future line of research could
rest upon linking morpho-topographic features of the glacier such has glacier surface area,
mean altitude or slope to the regression coefficients of both annual and seasonal SMB *vs.*



albedo relationships, giving the opportunity to establish analogy between monitored and
unmonitored glaciers. Tests have been carried out but no significant and satisfying results have
been obtained, due to a presumably too heterogeneous data set, where large glaciers (>10 km$^2$)
and/or south-facing glaciers are largely under-represented. Larger scale studies and multi-
variable correlations in between morpho-topographic features could be for instance envisaged.
Using the albedo method at a seasonal scale has shown promising results, especially for the
summer period where significant correlations have been found for the six seasonally
monitored glaciers. There is still in this approach a step to retrieve the seasonal SMB of an
unmonitored glacier with high confidence. The winter season has shown results that are for
now not entirely satisfactory. Glaciers that experience complete snow coverage during most of
the winter season showed the lowest correlation ($r^2 \leq 0.33$) while the two glaciers showing the
best correlations are subject to some events of surface melting in their lower reaches,
particularly at the end of the winter season. Therefore, studying the albedo signal in winter
could record snowfall events but seems to be little sensitive to snowfall intensity.
An additional approach has been carried out, aiming at retrieving $b_w$ by deduction from the
reconstructed $b_a$ and $b_s$ from the albedo signal. This approach, not using the winter albedo
signal, is poorly correlated ($r^2 < 0.16$) to in situ $b_w$ for the six seasonally monitored glaciers.
Indeed, the result extremely depends on the quality of the correlations between $b_a$, $b_s$ and the
albedo signals. Saint-Sorlin Glacier is a good example, being one of the glaciers with the
highest correlations for the annual ($r^2 = 0.86$) and summer ($r^2 = 0.94$) SMB. Subtracting $b_s$
from $b_a$ to computed $b_w$ leads to an average difference between computed and measured $b_w$ of
±0.41 m w.e for the 10 simulated years. As a consequence, in case of low correlations between
SMB and albedo, errors in the computed winter SMB become exacerbated.
**6   Conclusion**
In this study, we used the so-called albedo method to correlate annual and seasonal SMB to
glacier-wide average albedos obtained from MODIS images. This method has been carried on
30 glaciers located in the French Alps, over the period 2000-2015. Images processing has been





performed using the MODImLab algorithm, and filters on the images have been applied,
removing images with more than 30% cloud coverage, and excluding images with satellite
observation angles greater than 30°. Quality assessment has been performed and close
agreement has been found between albedos from AWS installed on Glacier de Saint-Sorlin
and MODIS retrieved albedo values. Annual SMB have been significantly correlated to the
summer minimum albedo for 27 of the 30 selected glaciers, confirming this variable as a good
proxy of the glacier-wide annual SMB. For the six seasonally monitored glaciers, summer
SMB obtained from the glaciological method have been significantly linked to the integral of
the summer albedo. For the winter season, implementing an albedo threshold for computing
the winter integral of the albedo has substantially improved the determination coefficients but
no uniform threshold has been found for the six selected glaciers. Two small glaciers, Saint-
Sorlin and Talèfre presented high correlation using albedo threshold, providing the
opportunity to reconstruct missing years or extending time series of these glaciers. Good
results have been obtained without using albedo thresholds in winter for Argentière and Mer
de Glace glaciers (>10 km$^2$) and further study would be required on a more exhaustive set of
large glaciers. We hence reconsider the idea proposed by Sirguey et al. (2016) of using albedo
thresholds to detect snow falls covering the glacier surface but albedo thresholds seem to
maximize artificially the correlation between winter SMB and winter integrated surface
albedo.
Sensitivity study on the impact of the considered cloud coverage has revealed a high
confidence in the MODImLab cloud algorithm, limiting pixel misclassifications, and a rather
high tolerance of the integrated signal to the number of partly cloud-covered images. This
confidence on cloud filters is very promising to document unmonitored glaciers. Correction of
shadows by the MODImLab algorithm has however revealed some limitations when a large
share of the glacier is shadowed by the surrounding topography (around winter solstice).
Despite this, glacier with severe and artificial drops of albedo in winter performed well when
quantifying the winter SMB (e.g. Argentière Glacier). Such systematic errors are therefore not





an issue for inter-annual studies, but would be a serious issue on studies focused on albedo
values themselves.
Using optical satellite images to estimate glacier surface processes and quantify annual and
seasonal SMB from the albedo cycle is therefore very promising and should be expanded to
further regions. Using images from different satellites, combining high spatial and temporal
resolution instruments, could substantially reduces uncertainties, especially for spotting the
albedo summer minimum with more confidence, but also to improve the temporal resolution.
This method could then in the short term, become reliable for retrieving SMB of monitored
and unmonitored glaciers.

**Acknowledgment**
This study was conducted within the *Service National d'Observation* GLACIOCLIM. Equipex
GEOSUD (*Investissements d'avenir* - ANR-10-EQPX-20) is acknowledged for providing the
2014 SPOT-6 images. The MODIS Level-1B data were processed by the MODIS Adaptive
Processing System (MODAPS) and the Goddard Distributed Active Archive Center (DAAC)
and are archived and distributed by the Goddard DAAC. In situ mass balance data for the
Glacier Blanc were kindly provided by the *Parc National des Ecrins*. The authors
acknowledge the contribution the Labex OSUG@2020 (*Investissements d'avenir* – ANR10
LABX56). Pascal Sirguey thanks the University of Grenoble Alpes and Grenoble-INP for the
6-month "invited professor grant" obtained in 2015-16.



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
