# Peer review of "annual surface mass balances from optical remote-sensing data Lucas Davaze1, Antoine Rabatel1, Yves Arnaud1, Pascal Sirguey2, Delphine Six1, Anne Letreguilly1, Marie Dumont3 1 Université Grenoble Alpes, CNRS, IRD"

_The Cryosphere, 2017_

## Short Comment (SC1) · 28 Apr 2017

For those interested in the supplementary material of this discussion paper, layout issue can occur for users using both Windows and Firefox. The problem is solved by downloading the supplementary material and opening it outside Firefox.

We're sorry for any inconvenience this might have caused.

---

## Referee Comment (RC1) · Anonymous Referee #1 · 23 May 2017

This manuscript is about a method to estimate annual and seasonal mass balances from relations received by using optical satellites on board of the platform TERRA with the sensor MODIS.

General comments:

This study was elaborated very carefully and it presents an interesting contribution to the estimation of surface mass balances for unmeasured glaciers. The paper is in general well written with a clear and logical structure. However, there are some basic points, which have to be discussed. Albedo is mainly a surface specific individual site

characteristic and it influences strongly the surface energy balance mainly through the shortwave incoming radiation. Therefore, it is clear that process based relations exist, which influence the melt behaviour of a glacier. This is clearly shown by the authors in the paper through the high correlations between summer albedo observations and summer surface mass balance. This part of the paper is excellent and should be kept as it is. However, the relations for winter mass balances the authors try to look for are not really obvious as they are not based on the same processes. Winter balance is mainly influenced by accumulation of solid precipitation and it is in whole paper not clearly stated why winter albedo should have a correlation to accumulation. With the MODIS sensor, it is possible to detect the albedo but never the amount of accumulation, which is the most important variable for winter surface mass balance. It is also clear and the authors mention that correctly in their paper that for larger glaciers (such as Mer de Glace and Glacier d'Argentière) the correlation of integrated albedo is increasing. This is obvious as these glaciers in general have a different behaviour of accumulation than smaller glaciers. Especially in maritime environments one have sometimes even during the accumulation period strongly varying albedo because lower parts of the glaciers show more variability concerning the change between melt and accumulation. Therefore, it can be assumed that the method developed by the authors probably work better in maritime areas than in continental ones. For the small glaciers they introduced a threshold albedo, which varies with each glacier. Thanks to this threshold, they found higher correlations for two of the small glaciers like Saint-Sorlin and Talefère. However, as they state by themselves in the conclusion section: '. . . but albedo thresholds seem to maximize artificially the correlation between winter SMB and winter integrated surface albedo'. Therefore, the approach is not really suited for the winter mass balance and it is strongly recommended that the authors remove the part related to the winter mass balance in their paper as it is not really related to any substantial processes, which they can capture with their method. It would make the paper much more concise, shorter and more reader friendly.

Specific comments:

Introduction: General: the authors are citing a lot of their own work. It would be appropriate to acknowledge also the international literature about albedo such as the work done by Klok, Knap, Painter, Pope and Takeuchi.

Line 47: add before the references e.g.

Line 49: this is not a very representative list of literature, please add more relevant literature here!

Line 51: Stocker 2013 is not a very good reference at this place of the introduction section. Please add relevant literature which is more specific for the content of your sentence. Line 55: Sort countries in brackets alphabetically

Line 57: replace 'little' by 'small'

Line 64: insert 'To reach this aim (maybe replace by objective) . . .

Line 81: However, this method is still the best one can do and your method is also based on calibration -> therefore this is a very weak reasoning for your method! Please delete this sentence!

Study area and data:

Line 115: but this is in fact a very strong reduction in the number of available glaciers and it reduces the representativeness of the application of the method. Therefore, it is not a very good argument against the study of Drolon et al 2016 mentioned on line 80

Line 154: why not using the individual values to estimate better the uncertainties?

Methods:

Line 177: Sentence misses a verb!

Line 230: Sentence is not clear, please clarify!

Line 237: What happens if you have strong summer snow fall events?

Line 276-277: is good but contradicts a little bit the objective mentioned in abstract and introduction at the beginning! You want to do it simple but then you agree that it is laborious?

Results:

Line 333: and MODIS does not see these areas?

Line 341: why not selecting only one or two MODIS Pixels in flat parts and observing and comparing these with the measurements and using then for evaluation?

Line 381: Two main observations stand . . .

Line 401: use m as unit instead of cm!

Line 430: However, this is important or not! I do not understand the justification within this chapter?

Discussion:

Line 626: such as instead of such has

Line 632: see study of Machguth, H., Haeberli, W., and Paul, F., 2012, Mass-balance parameters derived from a synthetic network of mass-balance glaciers: Journal of Glaciology, v. 58, no. 211, p. 965-979.
* * *

---

## Referee Comment (RC2) · J. Dozier (Referee) · 30 Jun 2017

First, I need to apologize for the delay in responding to the request for review. I can blame it on the NAS Decadal Survey for Earth Sciences and Applications programs, which has consumed me as I co-chair the Hydrology and Water Resources Panel.

A lot of work has gone into this paper, but the presentation of the results is not satisfactory. The measurement of albedo is rigorously and correctly done, following Dumont et al. (2012). However, the statistical correlation analysis of the minimum summer albedo

with the surface mass balance (SMB) sweeps a lot of interesting processes into it, and raises more questions. Specifically, how do the spatial and temporal variability of the albedo affect the SMB? And how does the uncertainty in the albedo estimate relate to the uncertain in the absorption, i.e. "1-$\alpha$" , which is what we really want.

Therefore, a much stronger paper would result if instead you compute the absorbed solar radiation by date, and integrate it over periods of interest. This calculation, perhaps called the radiative forcing in W m–2, takes into account the variability on the glacier, along with the different glaciers' exposure to the incoming solar radiation. Painter has used this measure. Painter and his colleagues have used this measure effectively to characterize the effects of dust on snow. You've already had to do much of the work in order to compute the albedo values, so take the next step.

I agree with Anonymous Referee #1, who questions the reliability of the analysis for the accumulation phase of the annual glacier cycle, which is likely driven more by the winter snowfall. After all, there are three ways to make a glacier shrink: darken it, starve it, or warm it. The analysis presented concentrates on the darkening. Tying this into the radiation balance would strengthen the contribution.

A nit of a comment concerns the statement on Lines 74-77. David Shean's work using photogrammetry from fine-resolution commercial sensors from DigitalGlobe shows much promise for interpreting successive DEMs to understand glacier shrinking.

Miller, S. D., F. Wang, A. B. Burgess, S. M. Skiles, M. Rogers, and T. H. Painter (2016), Satellite-based estimation of temporally resolved dust radiative forcing in snow cover, Journal of Hydrometeorology, 17, 1999-2011, doi: 10.1175/JHM-D-15-0150.1.

Shean, D. E., et al. (2016), An automated, open-source pipeline for mass production of digital elevation models (DEMs) from very-high-resolution commercial stereo satellite imagery, ISPRS Journal of Photogrammetry and Remote Sensing, 116, 101-117, doi: 10.1016/j.isprsjprs.2016.03.012.

---

## Author Comment (AC1) · 27 Jul 2017

**Response to reviewer #1**

**General comments:**

**Reviewer #1, comment #1:**

*However, the relations for winter mass balances the authors try to look for are not really obvious as they are not based on the same processes. Winter balance is mainly influenced by accumulation of solid precipitation and it is in whole paper not clearly stated why winter albedo should have a correlation to accumulation. With the MODIS sensor, it is possible to detect the albedo but never the amount of accumulation, which is the most important variable for winter surface mass balance. It is also clear and the authors mention that correctly in their paper that for larger glaciers (such as Mer de Glace and Glacier d'Argentière) the correlation of integrated albedo is increasing. This is obvious as these glaciers in general have a different behaviour of accumulation than smaller glaciers. Especially in maritime environments one have sometimes even during the accumulation period strongly varying albedo because lower parts of the glaciers show more variability concerning the change between melt and accumulation. Therefore, it can be assumed that the method developed by the authors probably work better in maritime areas than in continental ones. For the small glaciers they introduced a threshold albedo, which varies with each glacier. Thanks to this threshold, they found higher correlations for two of the small glaciers like Saint-Sorlin and Talefère. However, as they state by themselves in the conclusion section: '. . . but albedo thresholds seem to maximize artificially the correlation between winter SMB and winter integrated surface albedo'. Therefore, the approach is not really suited for the winter mass balance and it is strongly recommended that the authors remove the part related to the winter mass balance in their paper as it is not really related to any substantial processes, which they can capture with their method. It would make the paper much more concise, shorter and more reader friendly.*

**Authors reply:**

The potential of winter surface albedo to be a proxy of the winter surface mass balance (SMB) has been first introduced, with success, by Sirguey et al. (2016), using an albedo threshold to improve the correlation between winter integrated albedo and winter SMB. One of the conclusions of Sirguey et al. study was on the ability of such an approach to monitor the frequency of snowfall events, themselves proxy of the accumulation of snow on the glacier.

As you notify in your comment, one of the main conclusion of our study refutes the use of the winter integrated albedo as a proxy of the winter SMB using a constant albedo threshold at regional scale. Then, we agree with your suggestion to shorten the paper and will remove the part of the manuscript focusing on the relation between albedo and winter SMB:

-Method Sect. 3.3.2 will be modified by removing equations on winter detection.

-Section 4.3.3 will be removed.

-Section 5.3 will be removed and an assessment on the attempts/conclusions/perspectives on the winter season will be added in Sect. 5.4 (*"Limits of the albedo method"*). We will also add in this section that this method should not be definitively forsaken and as expressed in our conclusions, it can be used as a void filling method for time series with missing SMB data.

-Other sentences related to winter detection in the rest of the manuscript will also be removed.

We will also add in the conclusion that: *"monitoring winter glacier surface albedo may provide good insights on the frequency of snow accumulation at the surface of the glacier but lacks in quantifying the amount of accumulation."*

The comment on the importance of the size and type (maritime *vs.* continental) of studied glaciers is of a great interest and will be added to the manuscript as a hypothesis of the different results between Sirguey *et al.* (2016) and this study.

*"Another difference between our study and Sirguey et al., 2016, is that their work focused only on Brewster glacier, defined as maritime glacier. These types of glaciers, even during the accumulation period can experience strong varying albedos in their lower reaches, which lead to similar behaviors in winter as for Argentière and Mer de Glace glaciers. An interesting perspective would be to apply the method on a set of other maritime glaciers."*

Sirguey, P., Still, H., Cullen, N. J., Dumont, M., Arnaud, Y., & Conway, J. P. (2016). Reconstructing the mass balance of Brewster Glacier, New Zealand, using MODIS-derived glacier-wide albedo. *The Cryosphere*, *10*(5), 2465–2484. https://doi.org/10.5194/tc-10-2465-2016

**Specific comments:**

**Reviewer #1, comment #2:**

*Introduction: General: the authors are citing a lot of their own work. It would be appropriate to acknowledge also the international literature about albedo such as the work done by Klok, Knap, Painter, Pope and Takeuchi.*

**Authors reply:**

We will take into consideration the studies mentioned by Reviewer #1 and add them into the manuscript when justified. Other references will also be added to answer your comment #3, #4, #5 and one comment of Reviewer #2. Below is a list of references that will be added in the introduction of new version of the manuscript.

International literature about albedo will be added:

*Greuell, W. and Knap, W. H.: Remote sensing of the albedo and detection of the slush line on the Greenland ice sheet, J. Geophys. Res. Atmospheres, 105(D12), 15567–15576, 2000.*

*Greuell, W., Kohler, J., Obleitner, F., Glowacki, P., Melvold, K., Bernsen, E. and Oerlemans, J.: Assessment of interannual variations in the surface mass balance of 18 Svalbard glaciers from the Moderate Resolution Imaging Spectroradiometer/Terra albedo product, J. Geophys. Res. Atmospheres, 112, D07105/1null, 2007.*

*Shea, J. M., Menounos, B., Moore, R. D. and Tennant, C.: An approach to derive regional snow lines and glacier mass change from MODIS imagery, western North America, The Cryosphere, 7(2), 667–680, doi:10.5194/tc-7-667-2013, 2013.*

To expand the referenced study on the snow map method, this reference will be added to the manuscript:

*Chaponniere, A., Maisongrande, P., Duchemin, B., Hanich, L., Boulet, G., Escadafal, R. and Elouaddat, S.: A combined high and low spatial resolution approach for mapping snow covered areas in the Atlas mountains, Int. J. Remote Sens., 26(13), 2755–2777, 2005.*

As a complement of the quoted studies on the ELA method, following references will be added:

[revised manuscript text omitted]

**Reviewer #1, comment #5:**

Line 51: Stocker 2013 is not a very good reference at this place of the introduction section. Please add relevant literature which is more specific for the content of your sentence.

**Authors reply:**

We will also cite the work of *"Dyurgerov and Meier, 2000; Haeberli and Beniston, 1998; Oerlemans, 1994; Oerlemans, 2001"* for the line 51-54.

*Dyurgerov, M. B. and Meier, M. F.: Twentieth century climate change: Evidence from small glaciers, Proc. Natl. Acad. Sci., 97(4), 1406–1411, doi:10.1073/pnas.97.4.1406, 2000.*

*Haeberli, W. and Beniston, M.: Climate Change and Its Impacts on Glaciers and Permafrost in the Alps, Ambio, 27(4), 258–265, 1998.*

*Oerlemans: Glaciers and Climate Change, Balkema. [online] Available from: http://dspace.library.uu.nl/handle/1874/22045 (Accessed 13 July 2017), 2001.*

*Oerlemans, J.: Quantifying Global Warming from the Retreat of Glaciers, Science, 264(5156), 243–245, 1994.*

**Reviewer #1, comment #6:**

*Line 55: Sort countries in brackets alphabetically*

**Authors reply:**

Countries will be sorted alphabetically.

*"(e.g., France, Norway, Sweden, Switzerland)"*

**Reviewer #1, comment #7:**

*Line 57: replace 'little' by 'small'*

**Authors reply:**

Your remark will be considered.

*"However, this represents only a small sample of the nearly 250,000 inventoried glaciers ..."*

**Reviewer #1, comment #8:**

*Line 64: insert 'To reach this aim (maybe replace by objective) . . .*

**Authors reply:**

Indeed, this will be corrected.

*"To reach this objective, the development of ..."*

**Reviewer #1, comment #9:**

*Line 81: However, this method is still the best one can do and your method is also based on calibration -> therefore this is a very weak reasoning for your method! Please delete this sentence!*

**Authors reply:**

This sentence will be deleted in the new version of the manuscript.

**Reviewer #1, comment #10:**

*Line 115: but this is in fact a very strong reduction in the number of available glaciers and it reduces the representativeness of the application of the method. Therefore, it is not a very good argument against the study of Drolon et al 2016 mentioned on line 80*

**Authors reply:**

According to the comment #9, sentence line 81 will be removed.

The aim of this study is to validate the method of Sirguey *et al.* (2016) at both regional and seasonal scale before applying it to unmonitored glaciers. We agree that criteria (i) and (iv) are very reductive conditions but mandatory at a validation stage of the method. Critera (ii) could be relaxed by using a higher resolution sensor than MODIS and criteria (iii) is intrinsic at the detection method itself (observation of snow/ice evolution at the surface of a glacier between 0.459 and 14.385 µm).

*Studied glaciers have been selected following four criteria related to the availability of field data and remote sensing constraints, namely: (i) the annual glacier-wide SMB for the study period had to be available; (ii) the glacier surface area had to be wide enough to allow robust multi-pixel analysis; (iii) the glacier had to be predominantly free of debris to allow remotely-sensed observations of the albedo of snow and ice surfaces; and (iv) seasonal SMB records had to be available to consider seasonal*

*variability.*

Sirguey, P., Still, H., Cullen, N. J., Dumont, M., Arnaud, Y., & Conway, J. P. (2016). Reconstructing the mass balance of Brewster Glacier, New Zealand, using MODIS-derived glacier-wide albedo. *The Cryosphere*, *10*(5), 2465–2484. https://doi.org/10.5194/tc-10-2465-2016

**Reviewer #1, comment #11:**
*Line 154: why not using the individual values to estimate better the uncertainties?*
**Authors reply:**
We decided to average the available mass balance datasets together to be able to derive for each glacier a single relationship SMB - albedo. Finally, we did not assess the difference in between considered datasets because these differences have been investigated by Rabatel et al. (2016).
These sentences will be added in the new version of the manuscript:

"For the six glaciers where glacier-wide annual SMB are available from the two methods, i.e., in situ and satellite measurements, the average of the two estimates was used to calibrate and evaluate the albedo method, *in order to derive for each glacier a single relationship SMB vs. computed albedo. We did not assess the difference in between considered datasets because these differences have been investigated by Rabatel et al. (2016).*"

Rabatel, A., Dedieu, J. P. and Vincent, C.: Spatio-temporal changes in glacier-wide mass balance quantified by optical remote sensing on 30 glaciers in the French Alps for the period 1983–2014, J. Glaciol., 62(236), 1153–1166, doi:10.1017/jog.2016.113, 2016.

**Reviewer #1, comment #12:**
*Line 177: Sentence misses a verb!*
**Authors reply:**
We will add a verb to this sentence.
*"The MODImLab toolbox also produces sensor geometrical characteristics..."*

**Reviewer #1, comment #13:**
*Line 230: Sentence is not clear, please clarify!*
**Authors reply:**
This will be done as follow: *"Only computed $\alpha^{a}_{min}$ occuring in summer have been considered because minimum values out of summer are most likely artifacts."*

**Reviewer #1, comment #14:**
*Line 237: What happens if you have strong summer snow fall events?*
**Authors reply:**
As no albedo thresholds are used for the summer detection, integrated summer albedo accounts also for snowfall events (punctual high albedos). As an example, a strong summer snow fall event leading to a rather persistent snow coverage of the glacier would 'feed' the integrated albedo and will significantly reduces the glacier melting, which has an impact on the SMB (Oerlemans and Klok, 2004). The method therefore accounts for snowfall event to retrieve the glacier SMB.

This example will be added to the manuscript:

*Integrated summer albedo accounts also for snowfall events (punctual high albedos). As an example, a*

*strong summer snow fall event leading to a rather persistent snow coverage of the glacier would 'feed' the integrated albedo, and will physically reduces the glacier melting, which has an impact on the SMB (Oerlemans and Klok, 2004). The method therefore accounts for snowfall events to retrieve the glacier SMB.*

*Oerlemans, J., & Klok, E. J. (2004). Effect of summer snowfall on glacier mass balance. Annals of Glaciology, 38, 97–100. https://doi.org/10.3189/172756404781815158*

**Reviewer #1, comment #15:**

*Line 276-277: is good but contradicts a little bit the objective mentioned in abstract and introduction at the beginning! You want to do it simple but then you agree that it is laborious?*

**Authors reply:**

The word laborious will be changed as follows: *"This last step, although meticulous ..."*

**Reviewer #1, comment #16:**

*Line 333: and MODIS does not see these areas?*

**Authors reply:**

Yes MODIS sees these areas, but this method is only made to monitor the temporal evolution of glacier surface albedo. The MODImLab toolbox is designed to only calculate albedo over snow or ice covered area, because of brdf tables included within the toolbox (Dumont *et al.*, 2011; 2012). Monitoring extra glacial-feature would anyway introduce biased values in the glacier-wide albedo temporal evolution, even if the toolbox could have managed accurate extra-glacial surface albedo computing.

Dumont, M., Sirguey, P., Arnaud, Y., & Six, D. (2011). Monitoring spatial and temporal variations of surface albedo on Saint Sorlin Glacier (French Alps) using terrestrial photography. *The Cryosphere*, *5*(3), 759–771. https://doi.org/10.5194/tc-5-759-2011

Dumont, M., Gardelle, J., Sirguey, P., Guillot, A., Six, D., Rabatel, A., & Arnaud, Y. (2012). Linking glacier annual mass balance and glacier albedo retrieved from MODIS data. *The Cryosphere*, *6*(6), 1527–1539. https://doi.org/10.5194/tc-6-1527-2012

**Reviewer #1, comment #17:**

*Line 341: why not selecting only one or two MODIS Pixels in flat parts and observing and comparing these with the measurements and using then for evaluation?*

**Authors reply:**

This filtering part of the albedo is made to avoid detection of extra-glacial feature due to large sensor viewing angle. Comparison between single MODIS pixel and AWS measurement for validation has been performed in the Sect. 4.1 of the results.

**Reviewer #1, comment #18:**

*Line 381: Two main observations stand . . .*

**Authors reply:**

The sentence will be rewritten according to your remark as follows: *"Two main observations stand out from the winter part..."*

**Reviewer #1, comment #19:**

*Line 401: use m as unit instead of cm!*

**Authors reply:**

The unit *cm* will be replaced by *m* everywhere it is needed.

**Reviewer #1, comment #20:**

*Line 430: However, this is important or not! I do not understand the justification within this chapter?*

**Authors reply:**

According to your comment, this section will be re-formulated and simplified.

*"Looking at the 27 glaciers for which significant relationships have been found, 2001 is regularly identified as an outlier. According to existing SMB datasets, 2001 is the only year of the period 2000-2015 for which the annual SMB has been positive for all the studied glaciers (0.80 m w.e. yr$^{-1}$ in average).*

*To predict correctly the surface mass balance values for the year 2001 using the albedo method, monitored minimum glacier-wide average albedo would need to be extremely high (often greater than 0.7, i.e. 0.83 and 0.95 for Rochemelon and Vallonnet glaciers, respectively), to match the regression line derived from other years of the time series (Table 1). Taking into consideration snow metamorphism during the summer period, melting at the surface and possible deposition of debris or dusts, monitoring such high albedo values averaged at the glacier scale is unrealistic. As removing 2001 from the time series does not increase the number of glaciers for which the correlation is significant, 2001 has been conserved in the time series. However, this observation reveals a limitation of the albedo method by under-estimating the annual SMB value for years with very positive annual SMB. "*

**Reviewer #1, comment #21:**

*Line 626: such as instead of such has*

**Authors reply:**

The text will be corrected according to your remark: *"... linking morpho-topographic features of the glacier such as glacier surface area, ..."*

**Reviewer #1, comment #22:**

*Line 632: see study of Machguth, H., Haeberli, W., and Paul, F., 2012, Mass-balance parameter derived from a synthetic network of mass-balance glaciers: Journal of Glaciology, v. 58, no. 211, p. 965-979.*

**Authors reply:**

Very interesting study. However, according to your main comment, we will remove all sections that focus on the winter season and as a consequence, the sentence pointed out in this comment will be also removed.

**Response to reviewer #2: Dr. J. Dozier**

**General comments:**
**Reviewer #2, comment #1:**

*A lot of work has gone into this paper, but the presentation of the results is not satisfactory. The measurement of albedo is rigorously and correctly done, following Dumont et al. (2012). However, the statistical correlation analysis of the minimum summer albedo sweeps a lot of interesting processes into it, and raises more questions. Specifically, how do the spatial and temporal variability of the albedo affect the SMB? And how does the uncertainty in the albedo estimate relate to the uncertain in the absorption, i.e. "1-α" , which is what we really want.*

*Therefore, a much stronger paper would result if instead you compute the absorbed solar radiation by date, and integrate it over periods of interest. This calculation, perhaps called the radiative forcing in W m−2, takes into account the variability on the glacier, along with the different glaciers' exposure to the incoming solar radiation. Painter has used this measure. Painter and his colleagues have used this measure effectively to characterize the effects of dust on snow. You've already had to do much of the work in order to compute the albedo values, so take the next step.*

**Authors reply:**

Thank you for these suggestions. However, this study presents a validation (on several glaciers and different climate conditions) of a new method proposed by Sirguey *et al.* (2016) to derive both annual and seasonal surface mass balance (SMB) of individual glaciers at regional scale, by monitoring the albedo cycle of the surface of the glacier. Focusing on the spatial and temporal variability of the albedo, how it can affect the SMB or computing the surface energy balance, based on albedo measurement from MODIS is of a great interest but deserve a whole study by itself.

We will however add a sentence in the conclusion of the new manuscript, mentioning a possible future line of research with the datasets used in this study:

*"As a line of future researches, MODIS archive together with processed albedos with MODImLab could be used to compute an integrated daily absorbed solar radiation (Miller et al., 2016). This calculation could then be included in a surface energy balance computation, providing insights into the impact of the albedo variability on glacier SMB, especially in the melt season."*

Furthermore, at the light of your suggestion, we admit that the title of the study can be confusing and will be changed, in order to better represent the scope of the study: *"Monitoring glacier albedo as a proxy to derive summer and annual surface mass balances from optical remote-sensing data".*

Sirguey, P., Still, H., Cullen, N. J., Dumont, M., Arnaud, Y., & Conway, J. P. (2016). Reconstructing the mass balance of Brewster Glacier, New Zealand, using MODIS-derived glacier-wide albedo. *The Cryosphere*, *10*(5), 2465–2484. https://doi.org/10.5194/tc-10-2465-2016

*Miller, S. D., Wang, F., Burgess, A. B., McKenzie Skiles, S., Rogers, M. and Painter, T. H.: Satellite-Based Estimation of Temporally Resolved Dust Radiative Forcing in Snow Cover, J. Hydrometeorol., 17(7), 1999–2011, doi:10.1175/JHM-D-15-0150.1, 2016.*

**Reviewer #2, comment #2:**

*I agree with Anonymous Referee #1, who questions the reliability of the analysis for the accumulation phase of the annual glacier cycle, which is likely driven more by the winter snowfall. After all, there are three ways to make a glacier shrink: darken it, starve it, or warm it. The analysis presented concentrates on the darkening. Tying this into the radiation balance would strengthen the contribution.*

**Authors reply:**

According to Anonymous Referee #1 confirmed by your comment, we will removed the part of the study based on deriving winter SMB using the albedo proxy. The original purpose on focusing a part of the study on the relationship between albedo and winter SMB is given in the reply to Anonymous Referee #1 Comment #1.

As a complementary comment, the study does not focus on darkening but on the evolution of the glacier-wide albedo at the surface of the glacier, proxy of both melting events (lowering of the albedo because of a rising transient snowline) and accumulation events (abrupt rise of albedo after snowfall). We did not focus on monitoring albedo change due to dust or metamorphism, which could be a delicate topic according to the accuracy of sensors (Dumont *et al.*, 2014) even if some corrections are now processed within the new MODIS Collection 6 (Casey *et al.*, 2017) and as you suggested, recent studies have well performed such calculations on the impact of dusts on the snow albedo (Miller et al., 2016).

Casey, K. A., Polashenski, C. M., Chen, J. and Tedesco, M.: Impact of MODIS sensor calibration updates on Greenland ice sheet surface reflectance and albedo trends, Cryosphere Discuss, 2017, 1–24, doi:10.5194/tc-2017-38, 2017.

Dumont, M., Brun, E., Picard, G., Michou, M., Libois, Q., Petit, J.-R., Geyer, M., Morin, S. and Josse, B.: Contribution of light-absorbing impurities in snow to Greenland/'s darkening since 2009, Nat. Geosci., 7(7), 509–512, doi:10.1038/ngeo2180, 2014.

*Miller, S. D., Wang, F., Burgess, A. B., McKenzie Skiles, S., Rogers, M. and Painter, T. H.: Satellite-Based Estimation of Temporally Resolved Dust Radiative Forcing in Snow Cover, J. Hydrometeorol., 17(7), 1999–2011, doi:10.1175/JHM-D-15-0150.1, 2016.*

**Reviewer #2, comment #3:**

*A nit of a comment concerns the statement on Lines 74-77. David Shean's work using photogrammetry from fine-resolution commercial sensors from DigitalGlobe shows much promise for interpreting successive DEMs to understand glacier shrinking.*

**Authors reply:**

This reference will be updated. *"Several glacier surface properties have thus been used as proxies for volume fluctuations: changes in surface elevation from differencing digital elevation models (DEM) (e.g. Belart et al., 2017; Berthier et al., 2016; Gardelle et al., 2013; Ragettli et al., 2016; Shean et al., 2016)"*

*Belart, J. M. C., Berthier, E., Magnússon, E., Anderson, L. S., Pálsson, F., Thorsteinsson, T., Howat, I. M., Aðalgeirsdóttir, G., Jóhannesson, T. and Jarosch, A. H.: Winter mass balance of Drangajökull ice cap (NW Iceland) derived from satellite sub-meter stereo images, The Cryosphere, 11(3), 1501–1517, doi:10.5194/tc-11-1501-2017, 2017.*

*Berthier, E., Cabot, V., Vincent, C. and Six, D.: Decadal Region-Wide and Glacier-Wide Mass Balances Derived from Multi-Temporal ASTER Satellite Digital Elevation Models. Validation over the Mont-Blanc Area, Front. Earth Sci., 4, doi:10.3389/feart.2016.00063, 2016.*

*Gardelle, J., Berthier, E., Arnaud, Y. and Kääb, A.: Region-wide glacier mass balances over the Pamir-Karakoram-Himalaya during 1999–2011, The Cryosphere, 7(4), 1263–1286, doi:10.5194/tc-7-1263-2013, 2013.*

Ragettli, S., Bolch, T. and Pellicciotti, F.: Heterogeneous glacier thinning patterns over the last 40 years in Langtang Himal, Nepal, The Cryosphere, 10(5), 2075–2097, doi:10.5194/tc-10-2075-2016, 2016.

Shean, D. E., Alexandrov, O., Moratto, Z. M., Smith, B. E., Joughin, I. R., Porter, C. and Morin, P.: An automated, open-source pipeline for mass production of digital elevation models (DEMs) from very-high-resolution commercial stereo satellite imagery, ISPRS J. Photogramm. Remote Sens., 116, 101–117, doi:10.1016/j.isprsjprs.2016.03.012, 2016.

---

## Author Response (AR2)

Lucas Davaze

October 2017

IGE

rue Molière, Domaine Universitaire

38400 Saint Martin d'Hères, FRANCE

Tel: +33 7 82 94 15 97

lucas.davaze@univ-grenoble-alpes.fr

Authors response to reviewer.

Paper title: *Monitoring glacier albedo as a proxy to derive summer and annual surface mass balances from optical remote-sensing data*
Authors: L. Davaze, A. Rabatel, Y. Arnaud, P. Sirguey, D. Six, A. Letréguilly, M. Dumont
MS No.: https://doi.org/10.5194/tc-2017-49

Dear Handling Editor: Dr. Valentina Radic, dear reviewer: Dr Dozier,

We first thank you for taking the time to carefully read and comment the revised version of our paper.

According to the referee comments, we also would like to apologize for our short response in the previous review on the radiation balance computation, which was, retrospectively, lacking of argumentation.

Hereafter, we hope that you will find an exhaustive response to your comments (the text of the paper appears in blue and the changes we propose are underlined).

Best regards,

Lucas Davaze on behalf of all the authors.

**Response to review #2**

**General comments by V. Radic:**
*Your revised manuscript had been sent for another round of reviews since the reviewers asked for substantial revisions. By now I have received a review from Jeff Dozier and would like to give you a chance to respond to his comments and provide any further revisions if you consider necessary.*

*Jeff's opinion is that the revised manuscript is publishable, but he was disappointed that you refused his suggestion to compute a radiation balance. In your response letter please provide a convincing rationale for declining this suggestion.*

**As a reminder, below is the comment from the first review by Prof. J. Dozier :**
*"A lot of work has gone into this paper, but the presentation of the results is not satisfactory. The measurement of albedo is rigorously and correctly done, following Dumont et al. (2012). However, the statistical correlation analysis of the minimum summer albedo sweeps a lot of interesting processes into it, and raises more questions. Specifically, how do the spatial and temporal variability of the albedo affect the SMB? And how does the uncertainty in the albedo estimate relate to the uncertain in the absorption, i.e. "1-α" , which is what we really want.*
*Therefore, a much stronger paper would result if instead you compute the absorbed solar radiation by date, and integrate it over periods of interest. This calculation, perhaps called the radiative forcing in W m–2, takes into account the variability on the glacier, along with the different glaciers' exposure to the incoming solar radiation. Painter has used this measure. Painter and his colleagues have used this measure effectively to characterize the effects of dust on snow. You've already had to do much of the work in order to compute the albedo values, so take the next step."*

**Authors reply:**
We agree on the fact that snow albedo variation, especially through the darkening effect from dust deposition (Miller et al., 2016), and the intrinsic snow albedo feedback from snow grain growth due to warmer meteorological conditions (Warren, 1982), are important drivers of the snowpack melt. As stated in Prof Dozier comment, changes in glacier surface albedo have therefore a direct impact on the absorption of solar radiation by the snow/ice layers. Computing the absorbed solar radiation by date and for each glacier would then be an appropriated protocol to estimate the impact, in terms of snow or ice loss, of a changing surface albedo. However, our study is meant to propose a new method to derive summer and annual surface mass balance (SMB) of glaciers by analyzing the albedo signal from optical satellite images. It is beyond the scoop of this study to describe and quantify the impact of surface albedo changes on snow and ice melt; once again our goal is to present a methodological framework to compute annual and summer glacier SMB at regional scale with the aim of substantially increase the number of monitored glaciers.
We agree that computing the albedo at pixel scale of 250m resolution, for 30 glaciers offers the opportunity to study at a regional scale the impact of a changing surface albedo onto glacier

SMB, by studying the potential effect of increasing dust content, glacier orientation, or grain growth enhancement... and this will most probably be a next step of the work we can do with the data set of albedo maps presented in this study and data are available upon request from the corresponding author.

According to your remark, we added in the conclusion of the new manuscript an assessment of our discussion on the potential of computing a radiation balance:

[revised manuscript text omitted]